# SymmCD: Symmetry-Preserving Crystal Generation with Diffusion Models

**Daniel Levy**[*1,2], **Siba Smarak Panigrahi**[*1,2,3], **Sékou-Oumar Kaba**[*1,2],
**Qiang Zhu**[4], **Kin Long Kelvin Lee**[5], **Mikhail Galkin**[5],
**Santiago Miret**[5], **Siamak Ravanbakhsh**[1,2]
[1]McGill University, [2]Mila, [3]École Polytechnique Fédérale de Lausanne (EPFL),
[4]University of North Carolina at Charlotte, [5]Intel Labs

## Abstract

Generating novel crystalline materials has the potential to lead to advancements in fields such as electronics, energy storage, and catalysis. The defining characteristic of crystals is their symmetry, which plays a central role in determining their physical properties. However, existing crystal generation methods either fail to generate materials that display the symmetries of real-world crystals, or simply replicate the symmetry information from examples in a database. To address this limitation, we propose SymmCD[1], a novel diffusion-based generative model that explicitly incorporates crystallographic symmetry into the generative process. We decompose crystals into two components and learn their joint distribution through diffusion: 1) the asymmetric unit, the smallest subset of the crystal which can generate the whole crystal through symmetry transformations, and; 2) the symmetry transformations needed to be applied to each atom in the asymmetric unit. We also use a novel and interpretable representation for these transformations, enabling generalization across different crystallographic symmetry groups. We showcase the competitive performance of SymmCD on a subset of the Materials Project, obtaining diverse and valid crystals with realistic symmetries and predicted properties.

## 1 Introduction

Crystals serve as the fundamental building blocks of many materials, and the discovery of new crystalline materials is expected to lead to diverse technological breakthroughs in fields ranging from energy storage to computing hardware (Miret et al., 2024). Generative models have the potential to greatly accelerate this process by proposing new candidates materials, and possibly conditioning on desired properties or compositions.

The defining characteristic of crystals is their symmetry. These symmetries are Euclidean transformations that map the crystal structure back to itself. They can in general be some specific translations, rotations, reflections and combinations of these. The set of these operations is called the *space group* of the crystal. It is known that space groups in three dimensions fall into 230 distinct classes (Hahn et al., 1983). The symmetry of a crystal plays a crucial role in determining its stability along with its thermodynamic, electronic and mechanical properties (Nye, 1985). A classic example is given by piezoelectricity, the ability of a material to generate an electric dipole under mechanical stress, which can only be manifested in materials lacking inversion symmetry.

Importantly, many of the recently proposed generative models for crystals do not generate samples with non-trivial symmetry: for example, the most frequently generated crystals by DiffCSP (Jiao et al., 2023) and CDVAE (Xie et al., 2022) are in the low-symmetry P1 space group, which is very rare in nature. MatterGen (Zeni et al., 2023) can generate crystals conditioned on a desired space group for space groups that are highly represented in the dataset, but they only recover the target space group roughly 20% of the time, dropping to about 10% for more symmetric space groups.

---

[*]Equal Contribution. Correspondence to: `daniel.levy6@mail.mcgill.ca`
[1]Our code is publicly available at `https://github.com/sibasmarak/SymmCD/`

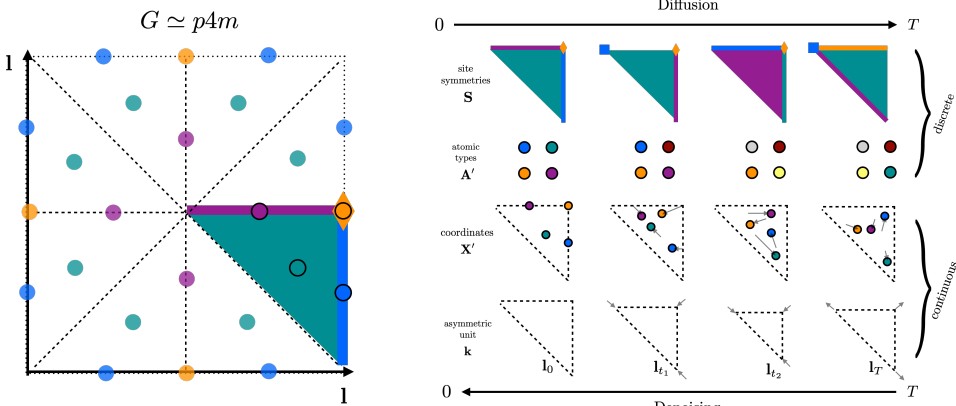

Figure 1: **Illustration of the SymmCD method.** *Left.* Representation of the unit cell of a 2D crystal with $p4m$ symmetry where the asymmetric unit and the site symmetries of the atoms are highlighted. Leveraging symmetry results in a much more compact, yet complete representation. *Right.* Diffusion and denoising on the different components of the representation. For site symmetries and atom types, discrete diffusion is used. For the coordinate and asymmetric unit continuous diffusion is used. The diffusion and denoising processes preserve the space group symmetry.

Cheetham & Seshadri (2024) analyse the space groups of the stable crystal structures proposed by the GNoME model of Merchant et al. (2023), finding that the top 4 most commonly generated space groups account for 34% of all generated crystals, even though each of those 4 space groups appears in less than 1% of crystals in the Inorganic Crystal Structure Database (Hellenbrandt, 2004).

In this work, we propose a novel approach for generative modeling of inorganic crystals that ensures any desired distribution of space groups. The idea is similar to that of creating a paper snowflake, where we fold the paper to create an unconstrained space, and after an unconstrained cutting of the paper in this space, its unfolding creates an object with desired symmetries. In the context of crystals, the unconstrained space is called the *asymmetric unit*, which is a maximal subset of the unit cell with no redundancy. To unfold the asymmetric unit, we need to generate the site symmetry of each atom inside the unit, i.e. the symmetry transformations that fix the atoms in place. In our generative process, the atomic positions are made consistent with generated site symmetries, enabling the unfolding of an asymmetric unit into a symmetric crystal; see Figure 1.

Crystals and their individual atoms have many different types of symmetries, so we need to address the issue of data-fragmentation. By representing symmetry information using standard crystallographic notations, such as Hermann–Mauguin notation or Wyckoff position labels (Hahn et al., 1983), we are faced with many crystals and site symmetries that have a low frequency in the training data. To address this problem, we introduce a novel representation of symmetries as binary matrices, which enables information-sharing and generalization across both crystal and site symmetries.

The main contributions of this work are as follows: **I)** We demonstrate a novel approach to generating crystals through the unconstrained generation of asymmetric units, along with their symmetry information. **II)** We introduce a physically-motivated representation for crystallographic site symmetries that generalizes across space groups. **III)** We experimentally evaluate our method, finding that it performs on par with previous methods at generating stable structures, while offering significantly improved computational efficiency due to our representation. **IV)** We analyse the symmetry and diversity of crystal structures generated by existing generative models.

## 2 RELATED WORK

There has been a growing body of work in developing machine-learning methods for crystal structure modeling, including the development of datasets and benchmarks (Jain et al., 2013; Saal et al., 2013; Chanussot et al., 2021; Miret et al., 2023; Lee et al., 2023; Choudhary et al., 2024). Recent work has also focused on developing architectures that are equivariant to various symmetries Duval et al. (2023) or are specifically designed to include inductive biases useful for crystal structures (Xie & Grossman, 2018; Kaba & Ravanbakhsh, 2022; Goodall et al., 2022; Yan et al., 2022; 2024).

In addition to structure-based modeling, prior work has also generated full-atom crystal structures, in which all atoms of the three-dimensional structure are generated. A range of generation methods including VAEs (Noh et al., 2019; Xie et al., 2022), GANs (Nouira et al., 2018; Kim et al., 2020), reinforcement learning (Govindarajan et al., 2023), diffusion models (Zeni et al., 2023; Yang et al., 2023; Jiao et al., 2023; Klipfel et al., 2024), flow-matching models (Miller et al., 2024), and active learning based discovery (Merchant et al., 2023) have been used. These follow similar works in 3D molecule generation (Hoogeboom et al., 2022; Garcia Satorras et al., 2021; Xu et al., 2022), but extend them by incorporating crystal periodicity. In addition to full-atom crystal generation, prior work has also applied text-based methods to understand and generate crystals using language models (Flam-Shepherd & Aspuru-Guzik, 2023; Gruver et al., 2024; Alampara et al., 2024).

Other works have pointed out the importance of symmetry of the generated structures. DiffCSP++ (Jiao et al., 2024), does so by using predefined structural templates from the training data and learning atomic types and coordinates compatible with the templates. While this is an interesting solution, we show that predefining the templates in this way severely limits the diversity and novelty of the generated samples. CrystalGFN (AI4Science et al., 2023) incorporates constraints on the lattice parameters and composition based on space groups but does not guarantee that the atomic positions respect the desired symmetry. Finally, the concurrent works CrystalFormer (Cao et al., 2024) and WyCryst (Zhu et al., 2024) generate symmetric crystals by predicting atom symmetries. However, they directly use the labels of Wyckoff positions to encode symmetries, which does not enable generalization across groups. The methods are therefore limited to generating from space groups that are common in the dataset. By contrast, our method generalizes across groups and can generate valid crystals even from groups that are rare in the dataset.

## 3 BACKGROUND

**Lattices and unit cells**   Crystals are macroscopic atomic systems characterized by a periodic structure. A crystal can be described as an infinite 3-dimensional *lattice* of identical *unit cells*, each containing atoms in set positions. We can represent a crystal with the tuple $\mathcal{C} = (\mathbf{L}, \mathbf{X}, \mathbf{A})$, where $\mathbf{L} = (\mathbf{l}_1, \mathbf{l}_2, \mathbf{l}_3) \in \mathbb{R}^{3 \times 3}$ is a matrix of *lattice vectors*, $\mathbf{X} \in [0, 1)^{3 \times N}$ represents the *fractional* coordinates of $N$ atoms within a unit cell, and $\mathbf{A} \in \{0, 1\}^{Z \times N}$ is a matrix of one-hot vectors of $Z$ possible elements for each atom. The lattice describes the tiling of unit cells: the cartesian coordinates of atoms can be given by $\mathbf{X}^c = \mathbf{L}\mathbf{X}$, and if $\mathbf{x}_i^c$ is the cartesian coordinate of an atom in a unit cell, then the crystal will also contain an identical atom at $\mathbf{x}_i^c + \mathbf{L}\boldsymbol{j}, \forall \boldsymbol{j} \in \mathbb{Z}^3$.

**Crystal symmetries**   In addition to the translational symmetry of the lattice, crystals typically have many other symmetries. Understanding these symmetries is fundamental in characterizing crystals and directly relates to many of the properties of these materials. The *space group $G$* of a crystal is the group of all Euclidean transformations that leave the crystal invariant, i.e., that simply permutes atoms of the same type. As space groups are subgroups of the Euclidean group, their elements can be represented as $(\mathbf{O}, \mathbf{t})$, where $\mathbf{O} \in O(n)$ and $\mathbf{t} \in \mathbb{R}^3$, with action on $\mathbf{x} \in \mathbb{R}^3$ defined as $(\mathbf{O}, \mathbf{t})\mathbf{x} = \mathbf{O}\mathbf{x} + \mathbf{t}$. The operations that are part of a space group can be generally understood as belonging to different types: translations, rotations, inversions, reflections, screw axes (combinations of rotations and translations), and glide planes (combinations of mirroring and translation).

Two space groups belong to the same *type* if all their operations can be mapped to each other by an orientation-preserving Euclidean transformation (coordinate change). We denote the set of all space group types as $\mathcal{G}$. In 3 dimensions, there are only 230 unique space group types. By choosing a canonical coordinate system, we can in general work only with space group types. The *point group $P$* of a space group $G$ is the image of the homomorphism $(\mathbf{O}, \mathbf{t}) \mapsto \mathbf{O}$, i.e the group obtained by keeping only the orthogonal parts of $G$. By contrast with space groups, any point group must at least preserve a single point, that is the origin. By a similar procedure to space groups, we can classify point groups and find that there are 32 crystallographic point group types, consisting of inversions, rotations, and reflections. We denote the set of all point group types as $\mathcal{P}$.

**Wyckoff positions**   Having classified symmetry groups, we can now also classify points of space using symmetry considerations. This will be important to our method, as we will seek to use these semantically meaningful classes to guide the generation process. Given a space group $G$, we say that two points $\mathbf{x}, \mathbf{x}' \in \mathbb{R}^3$ are part of the same *crystallographic orbit* if there is a $(\mathbf{O}, \mathbf{t}) \in G$ such that

$(\mathbf{O}, \mathbf{t}) \, \mathbf{x} = \mathbf{x}'$. The orbits form a partition of $\mathbb{R}^3$; they can be understood as the finest level of classification under $G$. We define the *site symmetry group* of a point $\mathbf{x}$, $S_\mathbf{x} = \{(\mathbf{O}, \mathbf{t}) \in G \mid (\mathbf{O}, \mathbf{t}) \, \mathbf{x} = \mathbf{x}\}$ as the subgroup of $G$ that leaves $\mathbf{x}$ invariant. It is clear that the site symmetry must be a point group (since translations do not preserve any point), and is a subgroup of $P$. From the orbit-stabilizer theorem (see e.g. Dummit & Foote (2004)), we can find that the number of points in the orbit $\mathbf{x}$ and in the unit cell is given by $|P|/|S_\mathbf{x}|$. Points in highly symmetric positions, therefore, result in smaller orbits. A point is in a *general position* if its site symmetry group is trivial. In this case, there is a one-to-one correspondence between points in the orbit and group members. If the site symmetry is non-trivial, a point is said to be in a *special position*.

Points in the same orbit have conjugate site symmetry groups. Therefore, site symmetry groups related by conjugation can be understood as equivalent. This motivates a coarser level of classification that will be very useful. Two points $\mathbf{x}, \mathbf{x}' \in \mathbb{R}^3$ are part of the same *Wyckoff position* if their site symmetry group is conjugate. Wyckoff positions have a clear meaning: they classify regions of space in terms of their type of symmetry. The *multiplicity* of a Wyckoff position is the number of equivalent atoms that must occupy that position and is equal to the $|P|/|S_\mathbf{x}|$ ratio introduced earlier.

**Asymmetric Units** The unit cell of a crystal can further be reduced into an *asymmetric unit*, which is a small part of the unit cell that contains no symmetry but can be used to generate the whole unit cell by applying the symmetry transformations of the space group. An asymmetric unit will only contain a single atom from each orbit.

## 4 METHOD: SYMMETRIC CRYSTAL DIFFUSION (SYMMCD)

### 4.1 REPRESENTATION OF CRYSTALS WITH WYCKOFF POSITIONS

As explained in the previous section, a crystal structure can in general be represented by the tuple $\mathcal{C} = (\mathbf{L}, \mathbf{X}, \mathbf{A})$. This representation has been used in previous generative models for crystals (Xie et al., 2022; Jiao et al., 2023; Luo et al., 2023; Zeni et al., 2023). However, a fundamental limitation of a model based on this representation is that it does not leverage the inductive bias of crystal symmetry and offers no guarantees for the crystal to satisfy anything but a trivial space group.

We introduce an alternative representation that respects symmetry in addition to having many desirable properties. First, we explicitly specify the space group type of the crystal $G \in \mathcal{G}$ in the representation. Given the space group, instead of representing each of the $N$ atoms individually with $\mathbf{X} \in \mathbb{R}^{3 \times N}$ and $\mathbf{A} \in \mathbb{R}^{Z \times N}$, we represent the $M$ crystallographic orbits; replicating the atoms within the orbit then creates the crystal. As explained in Section 3, the Wyckoff position identifies a set of orbits by site symmetries. Therefore, specifying the site symmetry and an arbitrary orbit representative is sufficient to identify a crystallographic orbit. This corresponds to a representation of an asymmetric unit within the unit cell. We thus define the set of orbit representatives with their Wyckoff positions as the tuple $\mathcal{C}' = (\mathbf{k}, \mathbf{X}', \mathbf{S}, \mathbf{A}')$, where $\mathbf{k}$ is a parametrization of the lattice (to be explained later), $\mathbf{X}' = [\mathbf{x}'_1, \dots, \mathbf{x}'_M] \in \mathbb{R}^{3 \times M}$ are the representative's fractional coordinates in the asymmetric unit, $\mathbf{S} = \left[ S_{\mathbf{x}'_1}, \dots, S_{\mathbf{x}'_M} \right] \in \mathcal{P}^M$ are the site symmetry groups and $\mathbf{A}' = [\mathbf{a}'_1, \dots, \mathbf{a}'_M] \in \mathbb{R}^{Z \times M}$ are the atomic types.

From the set of representatives, we can go back to the representation $\mathbf{X}$ and $\mathbf{A}$ in a unique way. This is done by generating the orbits using the replication operation that depends on the group $G$ and the site symmetry $\mathbf{S}$. The replication operation essentially consists of applying all of the symmetry operations of the space group except for the ones included in the site symmetry group. The details of this operation are included in Appendix A. Finally, the lattice $\mathbf{L}$ can be constrained to be compatible with the space group in a convenient way using the vector $\mathbf{k} \in \mathbb{R}^6$ (Jiao et al., 2024): $\log(\mathbf{L}) = \sum_i^6 k_i \mathbf{B}_i$, where the $\mathbf{B}_i \in \mathbb{R}^{3 \times 3}$ is a standard basis over symmetric matrices. This basis and the constraints on $\mathbf{k}$ for each space group are described in Appendix B.

Our representation of crystals that explicitly takes into account symmetry is therefore given by the tuple $\mathcal{C}' = (G, \mathbf{k}, \mathbf{X}', \mathbf{S}, \mathbf{A}')$. We convert crystal structures to this representation using the SPGLIB symmetry finding algorithm (Togo & Tanaka, 2018b) provided by PYMATGEN (Ong et al., 2013).

In addition to accounting for the symmetry, this representation of a crystal provides two important advantages compared to existing methods. First, it provides the generative model with a powerful

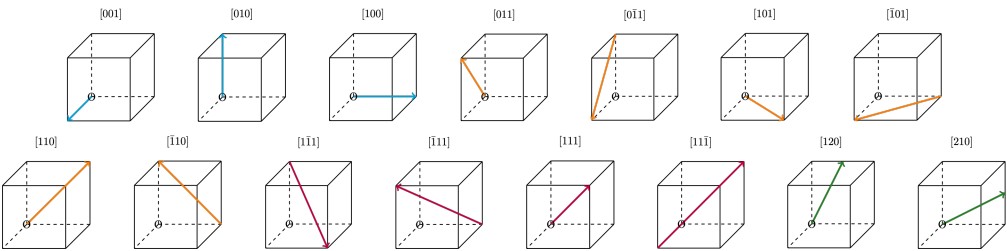

Figure 2: **Crystal symmetry axes.** The different axes describe the directions along which symmetry operations can occur. For each of the 15 axes, there are 13 possible symmetry operations.

physically-motivated inductive bias. It is known from crystallography that atoms are typically not located in arbitrary positions in the unit cell (Aroyo, 2013). Rather, it is energetically more favourable for atoms to occupy positions of high symmetry, e.g. special Wyckoff positions. The representation in terms of positions $\mathbf{X}$ does not make this explicit. The representation using Wyckoff positions $(\mathbf{X}', \mathbf{S}')$ provides explicit supervision to the model and guides the generation process: the model decides in which *type* of high-symmetry position an atom should be located and generates a position compatible with that type. Second, the representation in terms of Wyckoff positions is much more compact than the representation that operates on individual atoms. $M$ is often significantly smaller than $N$. In the MP-20 dataset (a subset of the Materials Project dataset (Jain et al., 2013)) for example, the average number of orbits is $\bar{M} = 4.7$ whereas the average number of atoms per unit cells is $\bar{N} = 18.9$, representing a fourfold difference [2]. We therefore eliminate the redundant information from the representation and increase the computational efficiency of our method.

## 4.2 SYMMETRY REPRESENTATION

A key component of our representation of crystals with Wyckoff positions is the encoding of the space group $G$ and site symmetry groups $\mathbf{S}'$. While there are many existing methods to encode these symmetries, they generally do not make explicit the commonalities between the site symmetries of Wyckoff positions in the same space group, and the commonalities between different space groups across crystal systems. This is an important limitation: because there are 230 space groups, not having a representation that is common across space groups results in dividing the effective amount of data the model is trained on by a large amount. For example, in the MP-20 dataset (Jain et al., 2013; Xie & Grossman, 2018) 113 space groups out of 169 in the training set have fewer than a hundred samples associated with them, and specific Wyckoff positions in each group have even fewer samples. We propose a method to represent the site symmetries of different Wyckoff positions and to encode the symmetries of space groups to address this shortcoming.

We represent atom site symmetries using a binary representation based on the oriented site symmetry symbol used by the International Tables for Crystallography to describe Wyckoff positions (Hahn et al., 1983; Donnay & Turrell, 1974). The oriented site symmetry symbols denote generators of the site symmetry group along different possible axes, illustrated in Figure 2. In total, there are 15 possible axes of symmetry in a crystal, corresponding to each of the Cartesian axes, along with body and face diagonals.

Examples of possible symmetry operations along each axis include rotations and roto-inversions, as well as mirror symmetry along a plane perpendicular to the axis. There are 13 possible symmetries along each axis. Listing out the site symmetry operation along each axis yields a $15 \times 13$ binary matrix, or equivalently 15 different one-hot vectors. There is an injective mapping between site symmetries and site symmetry matrix representations, so a representative atom can be replicated to produce a full orbit using this representation.

The space group $G$ can also be encoded into a binary representation using a similar scheme, by listing out the 15 possible axes of symmetry and listing out the possible symmetry operations along each axis. Unlike the point group symmetries of atoms, these space group symmetry operations may involve translations and so include screw and glide transformations, leading to 26 possible symmetry operations. Further details are included in Appendix 4.1. As a part of this project, both

---

[2]This is using the conventional unit cell, not the primitive unit cell. A conventional cell may be twice or four times as big as a primitive cell.

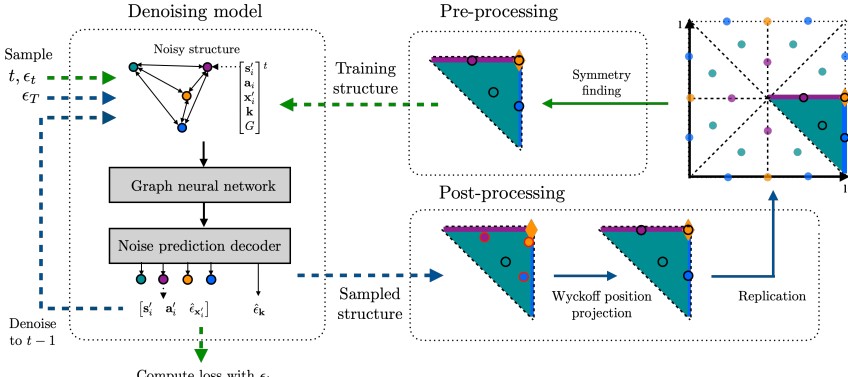

Figure 3: **SymmCD training and sampling pipeline.** For training (green), the crystal structures are pre-processed to find the space group $G$ along with the site symmetries $\mathbf{S}$ and a set of orbit representatives inside the asymmetric unit. The denoising model is a GNN with fully connected graphs, followed by a decoder. For sampling (blue), the positions are projected to the closest one compatible with their site symmetry. Then, the asymmetric unit is replicated to obtain the unit cell.

the site symmetry representations and the space group representations have been incorporated as functions in the PyXtal software library (Fredericks et al., 2021).

### 4.3 DIFFUSION MODEL

We can now describe the generative model and training process. In SymmCD, the space group and the number of orbit representatives are first sampled from separate distributions obtained from data, such that the distribution over crystal structures is $p\left(\mathcal{C}\right) = p\left(\mathbf{k}, \mathbf{X}', \mathbf{S}, \mathbf{A}' \mid M, G\right) p\left(M \mid G\right) p\left(G\right)$. We will seek to model the conditional distribution $p\left(\mathbf{k}, \mathbf{X}', \mathbf{S}, \mathbf{A}' \mid M, G\right)$ with a denoising diffusion model (Sohl-Dickstein et al., 2015; Ho et al., 2020).

We leverage our binary representation for incorporating crystal symmetry information (described in Section 4.2) and perform joint diffusion over lattice representation ($\mathbf{k}$), fractional coordinates of atoms ($\mathbf{X}'$), their types ($\mathbf{A}'$), and the associated binary representation of site symmetry ($\mathbf{S}$).

**Diffusion process** We consider a separate diffusion process over the different components of the crystal representation. We apply discrete diffusion from Austin et al. (2021) for site symmetries and atom types. Rather than adding Gaussian noise as in conventional diffusion, we add noise to categorical features by multiplying probability vectors by a transition matrix and sampling from the new probabilities. Inspired by Vignac et al. (2023), the transition matrices are parameterized so that the process converges to the marginals from the data distribution for atom types and site symmetries. The loss function used for discrete diffusion on atomic types is

$$\mathcal{L}_{\mathbf{A}'} = \mathbb{E}_{\mathbf{a}_t \sim \mathrm{Cat}(\mathbf{a}_0^\top \bar{\mathbf{Q}}_t), t \sim \mathrm{U}(1,T)} \sum_{i=1}^{M} \mathrm{CrossEntropy}(\mathbf{a}_i, \hat{\mathbf{a}}_i), \tag{1}$$

where $\mathbf{a}_0$ is the initial one-hot encoding of the atom types for a single representative and $\bar{\mathbf{Q}}_t = \prod_{i=1}^{t} \mathbf{Q}_i \in \mathbb{R}^{Z \times Z}$ is the cumulative product of transition matrices between timesteps, and $\hat{\mathbf{a}}_i$ are the predicted denoised probabilities. The same loss function is used for site symmetries.

Continuous diffusion is used for fractional coordinates and lattice parameters, similar to Jiao et al. (2023). The loss function for the continuous diffusion on lattice parameters is

$$\mathcal{L}_{\mathbf{k}} = \mathbb{E}_{\epsilon_{\mathbf{k}} \sim \mathcal{N}(0, \mathbf{I}), t \sim \mathrm{U}(1,T)} \left[||m \odot \epsilon_{\mathbf{k}} - \hat{\epsilon}_{\mathbf{k}}(\mathcal{C}'_t, t)||_2^2\right],$$

where $m$ is a space group-dependent mask, and $\hat{\epsilon}_{\mathbf{k}}$ is the predicted denoising vector. The same loss function is used for the fractional coordinates, except that to capture their periodic nature, we use a wrapped normal distribution $\mathcal{WN}(0, 1)^{3 \times M}$. We provide more details in Appendix D.

**Denoising network** The architecture of the denoiser is a message-passing neural network that operates on a fully connected graph of representatives, based on Jiao et al. (2023). Features for each representative $\mathbf{h}_i$ are initialized using an embedding of their atom types $\mathbf{a}_i$ and their site symmetries

$\mathbf{S}_i$, along with the graph-level features of the diffusion timestep $t$, the lattice features $\mathbf{k}$, and an embedding of the space group $G$. At each layer, messages $\mathbf{m}_{ij}$ are computed between representatives $i$ and $j$ by applying an MLP to $\mathbf{h}_i, \mathbf{h}_j$, and a Fourier embedding of the vector $\mathbf{x}_i - \mathbf{x}_j$ to respect periodic invariance. These messages are then used to update $\mathbf{h}_i$. More details on the architecture are included in Appendix E.1. Note that the denoising network is not equivariant. It is not necessary since the unit cell axes provide a canonical reference system (Kaba et al., 2023). We also found that using an equivariant denoising network like EGNN Satorras et al. (2021) did not work well in part due to the fact that since we use periodic encodings, the crystal structure input has a translational symmetry. An equivariant model may not be able to break that symmetry (Kaba & Ravanbakhsh, 2023) resulting in a inability to output correct positions in the asymmetric unit (or unit cell).

**Putting it all together** The algorithm for training our diffusion model is outlined in Algorithm 1. We use different loss coefficients $\lambda_\mathbf{k}, \lambda_{\mathbf{X}'}, \lambda_{\mathbf{A}'}$ and $\lambda_\mathbf{S}$ to weigh the importance of the different components of the model. The algorithm for sampling from the diffusion model is shown in Algorithm 2. The full pipeline is summarized in Figure 3. Since both the diffusion and the denoising process both operate only on the asymmetric unit, they fully preserve the symmetry of the crystal.

---

**Algorithm 1** Training SymmCD

1: **Input:** Dataset of crystals $\mathcal{D}$
2: **while** not converged **do**
3:      Sample a crystal $\mathcal{C} = (\mathbf{L}, \mathbf{X}, \mathbf{A})$ from dataset $\mathcal{D}$, and a timestep $t \sim \text{Uniform}(1, T)$
4:      Derive the asymmetric representation $\mathcal{C}' = (G, \mathbf{k}, \mathbf{X}', \mathbf{A}', \mathbf{S})$ from $\mathcal{C}$
5:      Add noise to $\mathbf{k}, \mathbf{X}', \mathbf{A}'$, and $\mathbf{S}'$:
6:         $\mathbf{k}_t = \sqrt{\bar{\alpha}_t}\mathbf{k}_0 + \sqrt{1 - \bar{\alpha}_t}\epsilon_\mathbf{k}, \quad \epsilon_\mathbf{k} \sim \mathcal{N}(0, \mathbf{I})$
7:         $\mathbf{X}'_t = \sqrt{\bar{\alpha}_t}\mathbf{X}'_0 + \sqrt{1 - \bar{\alpha}_t}\epsilon_{\mathbf{X}'}, \quad \epsilon_{\mathbf{X}'} \sim \mathcal{WN}(0, \mathbf{I})$
8:         $\mathbf{A}'_t \sim \text{Cat}(\mathbf{A}\bar{\mathbf{Q}}_{a,t})$
9:         $\mathbf{S}_{u,t} \sim \text{Cat}(\mathbf{S}\bar{\mathbf{Q}}_{u,G,t})$
10:     Use denoising network $\phi$ to predict $\hat{\epsilon}_\mathbf{k}, \hat{\epsilon}_{\mathbf{X}'}, \hat{\mathbf{A}}', \hat{\mathbf{S}}$ from noisy $\mathcal{C}_t = (G, \mathbf{k}_t, \mathbf{X}'_t, \mathbf{A}'_t, \mathbf{S}_t), t$
11:     Compute losses $\mathcal{L}_\mathbf{k}, \mathcal{L}_{\mathbf{X}'}, \mathcal{L}_{\mathbf{A}'}, \mathcal{L}_{\mathbf{S}'}$
12:     Update the denoising network $\phi$ using total loss:
13:        $\mathcal{L} = \lambda_\mathbf{k}\mathcal{L}_\mathbf{k} + \lambda_{\mathbf{X}'}\mathcal{L}_{\mathbf{X}'} + \lambda_{\mathbf{A}'}\mathcal{L}_{\mathbf{A}'} + \lambda_\mathbf{S}\mathcal{L}_{\mathbf{S}'}$
14: **end while**

---

**Algorithm 2** Sampling from SymmCD

1: **Input:** Target space group $G$, Number of representatives $M$
2: **Initialize:**
3:     Sample $\mathbf{k}_T \sim \mathcal{N}(0, \mathbf{I})$
4:     Sample $\mathbf{X}'_T \sim \mathcal{U}(0, 1)^{3 \times M}$
5:     Sample $\mathbf{A}'_T \sim p_{\text{marginal}}(\mathbf{A}')$
6:     Sample $\mathbf{S}'_T \sim p_{\text{marginal}}(\mathbf{S}'|G)$ (site symmetries)
7: **for** $t = T$ to $1$ **do**
8:     Compute $\hat{\epsilon}_\mathbf{k}, \hat{\epsilon}_{\mathbf{X}'}, \hat{\mathbf{A}}', \hat{\mathbf{S}}$ using denoising network $\phi(\cdot)$
9:     Sample $\mathbf{k}_{t-1}, \mathbf{X}'_{t-1}, \mathbf{A}'_{t-1}, \mathbf{S}'_{t-1}$ using $\hat{\epsilon}_\mathbf{k}, \hat{\epsilon}_{\mathbf{X}'}, \hat{\mathbf{A}}', \hat{\mathbf{S}}$.
10: **end for**
11: Project $\mathbf{S}'_0$ onto nearest valid point group
12: Project $\mathbf{X}'_0$ onto nearest Wyckoff position with that site symmetry
13: Replicate representative atoms $\mathbf{X}'_0$ using site symmetries $\mathbf{S}'_0$ to generate full crystal $\mathbf{X}_0$
14: **Output:** Crystal structure $\mathbf{X}_0$, Atom types $\mathbf{A}_0$, lattice $\mathbf{L}_0$

---

## 5 EXPERIMENTS

We test our model on *de novo* crystal generation using the MP-20 dataset (Xie et al., 2022), a subset of the Materials Project (Jain et al., 2013) consisting of 40,476 crystals, each with up to 20 atoms per primitive unit cell. The data is preprocessed to use the conventional unit cell rather than the primitive unit cell, as the former has more conveniently expressed symmetries and constraints. A conventional unit cell may be larger than a primitive unit cell, which results in up to 80 atoms in the unit cell. We withhold 20% of the dataset as a validation set, and 20% as a test set.

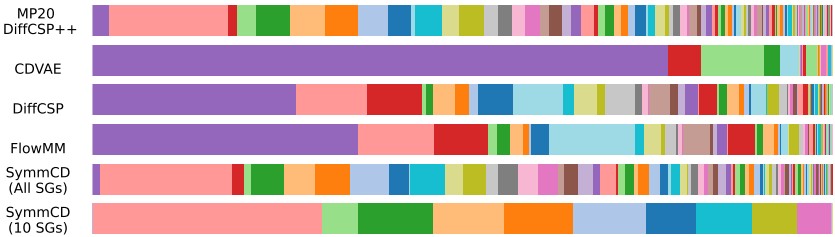

Figure 4: Proportion of space group symmetries of the dataset, and each method. The width of each color segment represents the proportion of crystals with that symmetry. From left to right, the first few space groups are: P1, Fm$\bar{3}$m, Cm, P$\bar{1}$, C2/m, I4/mmm, Pm$\bar{3}$m, P6$_3$/mmc, and Pm.

We empirically demonstrate our contributions, particularly in ensuring we generate crystals with desired symmetries while being competitive with existing baselines. We compare our proposed method with four recent strong baselines: CDVAE (Xie et al., 2022), DiffCSP (Jiao et al., 2023), DiffCSP++ (Jiao et al., 2024) and FlowMM (Miller et al., 2024). We retrained each method according to their given hyperparameters, and generated 10,000 crystals each. We also consider a variant of our model (10 SGs) where we only sample from the 10 most common space groups in the MP-20 dataset, similar to (Cao et al., 2024) [3]. This is to provide a more nuanced comparison with other methods, which are not constrained in matching the space group distribution. This choice still captures a large portion of the data distribution, since these are the most prevalent space groups.

## 5.1 SYMMETRY AND STRUCTURAL DIVERSITY

First, we evaluate the different methods on their ability to generate crystals with diverse structures and space groups. This aspect has not been investigated yet for the considered baselines, yet it is significant in understanding if they generate realistic structures.

**Space groups** To detect the space group of the generated structures, we use spglib's symmetry finding method (Togo & Tanaka, 2018a; Ong et al., 2013) with a tolerance of 0.1Å. The distribution of space groups of the generated structures is shown in Figure 4. It can be observed that while SymmCD matches the highly diverse data distribution, CDVAE mostly generates crystals with trivial $P1$ symmetry, and DiffCSP and FlowMM generate many crystals with low symmetry and generally have lower diver-

Table 1: Template statistics for each model.

| Method | # Unique | % in Train | # New |
|---|---|---|---|
| Training Set | 3318 | 100 | - |
| CDVAE | 797 | **28.7** | 568 |
| DiffCSP | 1347 | 43.2 | 764 |
| DiffCSP++ | 1905 | 94.2 | 110 |
| FlowMM | 1291 | 41.7 | 753 |
| SymmCD | **2794** | 40.8 | **1654** |

sity of space groups. We also consider a new quantitative metric to characterize the space group distribution, $d_{sg}$, which is calculated as the Jensen-Shannon distance between the distribution of space groups of the generated structures and the test set. We report it for the different methods in the rightmost column of Table 3. The results confirm that SymmCD and DiffCSP++ are the only methods that accurately match the distribution of space groups in the dataset.

**Unique Templates** We also evaluate the ability of the different methods to generate diverse crystal structures. We define a structural *template* to be a combination of a space group and a multiset of occupied Wyckoff positions, regardless of the atomic types in the Wyckoff position. Templates, also known as Wyckoff sequences, are used in practice to classify crystals by their symmetry. They have the advantage of providing a notion of a structure that is highly flexible, while being robust to perturbations of coordinates that do not change the position of atoms with respect to symmetry elements. Most potential templates have not yet been experimentally observed, motivating the development of methods that can discover materials with new templates (Hornfeck, 2022).

The training dataset contains 3318 such unique templates. We examine the templates for the 10,000 crystals generated by each method, and report results in Table 1. We find that SymmCD performs best out of all models, proposing unique and novel templates. This highlights an important limitation of DiffCSP++. While it can produce diverse space groups and to a certain extent diverse templates, since it uses pre-defined templates it fails to generate structures with *novel* templates. Our method does not suffer from this problem since it learns to generate templates.

---

[3] These space groups are numbered: 2, 12, 14, 62, 63, 139, 166, 194, 221, 225

## 5.2 STABLE, UNIQUE AND NOVEL (S.U.N.) STRUCTURES

Regardless of their target application, generative models for crystals should produce sets of crystals that are thermodynamically stable, unique (not duplicated within the predicted set), and novel (not already in the training data), or S.U.N. To this end, we adapt the evaluation procedure of Miller et al. (2024) to assess the capability of our model to generate S.U.N. materials. Thermodynamic stability is determined by estimating the energy of a material with respect to a *convex hull*. The convex hull gives linear combinations of known phases that represent the lowest-energy mixtures of materials;

Table 2: Percent of stable and S.U.N. samples produced from an initial set of 10,000 generated crystals for each method.

|                   | Initial Stable | Relaxed Stable | Relaxed S.U.N. |
| ----------------- | -------------- | -------------- | -------------- |
| CDVAE             | 0.24%          | 4.42%          | 4.26%          |
| DiffCSP           | 7.82%          | 11.32%         | 8.92%          |
| DiffCSP++         | 6.99%          | 11.36%         | 8.62%          |
| FlowMM            | 4.26%          | 9.05%          | 6.49%          |
| SymmCD (All SGs)  | 4.96%          | 9.34%          | 6.86%          |
| SymmCD (10 SGs)   | 6.85%          | 10.92%         | 7.59%          |

if a material has an energy above the hull, it is energetically favorable for it to decompose into a combination of stable phases and is therefore thermodynamically unstable. We assess the stability of generated crystals by estimating their energies using a pretrained CHGNet model (Deng et al., 2023), and comparing that to a convex hull computed for Materials Project (Riebesell et al., 2024).

We predict the stability of the 10,000 samples generated by each method, and then use CHGNet to compute relaxed structures for each crystal, which results in higher stability. Finally, we check whether the stable relaxed crystals are also unique and novel. Details of this procedure are included in Miller et al. (2024). The results are shown in Table 2. We see that SymmCD performs on par with the baselines. Sampling over the most common space groups results in more stable structures.

## 5.3 VALIDATION WITH DENSITY FUNCTIONAL THEORY RELAXATIONS

We carried out electronic structure calculations as a more accurate way to evaluate the crystal structures generated by the different methods. Since these computations are significantly more expensive than the ones with CHGNet, we performed them on 100 structures sampled from each model. Concretely, these involved using the `CP2K` (Kühne et al., 2020) suite of programs to perform cell and geometry relaxations to assess the quality of generated structures based on their distances from "true" local minima. We expect a better method to produce structures with less distance traveled, smaller per-atom forces (i.e. energy gradient with respect to positions), and with fewer iterations.

In these calculations, we find that the convergence rates of methods are not significantly different (see Figure 10). Comparing expectation values of the maximum force after optimization, however, we see that SymmCD is significantly more successful in generating structures that are readily optimized. The full results of this analysis, as well as the parameters used for CP2K can be found in Appendix F.6.

## 5.4 PROXY METRICS

Although ultimately we care about the stability, novelty, and properties of crystals after structural relaxation, we also compare the different methods using the proxy metrics established by Xie et al. (2022), as they are cheap enough to compute over a large set of generated samples and demonstrate the ability of different methods to capture the properties of a target distribution. These metrics include heuristics of validity, coverage of the test set, and the Wasserstein distances between the distributions of three properties of the generated samples and the test set: atomic densities $d_\rho$, number of unique elements $d_{elem}$, and predicted formation energy $d_E$. We also include $d_{sg}$, the Jensen-Shannon distance between space group distributions. Further details are included in Appendix F.5.

For each method, we retrained a model with 5 different seeds, generating 10,000 samples for evaluation per seed. The results are shown in Table 3. Although SymmCD performs similarly to other methods for most metrics, it is more likely to generate structurally invalid crystals, possibly because it cannot easily see distances between atoms. We also note the large variance in the results across different seeds for each method.

Table 3: Results for comparing the validity, coverage, and property distribution metrics.

| MP-20 | Validity (%) (↑) | | Coverage (%) (↑) | | Property Distribution (↓) | | | |
|---|---|---|---|---|---|---|---|---|
| | Struct. | Comp. | Recall | Precision | $d_\rho$ | $d_E$ | $d_{elem}$ | $d_{sg}$ |
| CDVAE | $99.97_{\pm0.03}$ | $85.61_{\pm1.20}$ | $99.31_{\pm.06}$ | $99.47_{\pm0.19}$ | $0.70_{\pm.10}$ | $0.24_{\pm.06}$ | $1.28_{\pm.06}$ | $0.69_{\pm.007}$ |
| DiffCSP | $97.43_{\pm3.10}$ | $82.50_{\pm1.34}$ | $99.55_{\pm.08}$ | $98.73_{\pm1.34}$ | $0.18_{\pm.08}$ | $0.14_{\pm.05}$ | $0.56_{\pm.07}$ | $0.44_{\pm.009}$ |
| DiffCSP++ | $99.44_{\pm0.06}$ | $86.50_{\pm0.85}$ | $99.72_{\pm.06}$ | $99.61_{\pm0.08}$ | $0.12_{\pm.04}$ | $0.05_{\pm.01}$ | $0.33_{\pm.04}$ | $0.16_{\pm.009}$ |
| FlowMM | $96.67_{\pm0.57}$ | $83.25_{\pm0.13}$ | $99.49_{\pm.05}$ | $99.58_{\pm0.10}$ | $0.23_{\pm.10}$ | $0.09_{\pm.02}$ | $0.08_{\pm.02}$ | $0.50_{\pm.011}$ |
| SymmCD$_{All SGs}$ | $90.34_{\pm3.39}$ | $85.81_{\pm0.87}$ | $99.58_{\pm.08}$ | $97.76_{\pm0.85}$ | $0.23_{\pm.06}$ | $0.21_{\pm.04}$ | $0.40_{\pm.06}$ | $0.164_{\pm.003}$ |
| SymmCD$_{10 SGs}$ | $92.30_{\pm6.10}$ | $87.13_{\pm0.62}$ | $97.33_{\pm.59}$ | $98.78_{\pm0.69}$ | $0.53_{\pm.23}$ | $0.21_{\pm.09}$ | $0.16_{\pm.02}$ | $0.469_{\pm.002}$ |

## 5.5 COMPUTATIONAL EFFICIENCY

We demonstrate significant computational efficiency gains and reduced memory footprint due to using a more compact representation based on crystallographic orbits. We compare our model to an equivalent model that looks at a full unit cell, rather than just the asymmetric unit. It also uses a fully connected graph to represent the atoms in the unit cell, but does not predict site symmetry representations. This makes the model essentially equivalent to DiffCSP (Jiao et al., 2023), but with the same architecture and hyperparameters as SymmCD for consistent comparison. We compare the two representations for one epoch of training using 40GB of RAM and a single NVIDIA MIG A100 and report the results in Table 4. These results highlight SymmCD's memory efficiency and faster training capabilities.

Table 4: Computational efficiency of our representation

| | Asymmetric Unit (ours) | Conventional Unit Cell |
|---|---|---|
| Maximum batch size (↑) | **8192** | 512 |
| Memory for 512 batch size (↓) | **3.6 GB** | 31 GB |
| Time for 1 training epoch (↓) | **27 sec.** | 52 sec. |

## 5.6 SCALING UP TO MPTS-52

To validate that SymmCD can scale up to datasets of larger crystals thanks to its compact representation, we also validate our model by training it on the more challenging MPTS-52 dataset (Baird et al., 2024), which contains crystals with up to 52 atoms per primitive unit cell, and includes more chemical elements. Further details of this dataset along with the results are included in Appendix F.1.

We note that none of the baselines we compare against (CDVAE, DiffCSP, FlowMM, and DiffCSP++) present results for de-novo generation on this dataset. We can see from these results that SymmCD is able to scale to much larger crystals than those in the MP-20 dataset and is still capable of producing valid, stable, and novel crystals that match the training distribution of the dataset. The metrics reported are worse than those for MP-20, as this is a more difficult dataset.

## CONCLUSION

In this paper, we introduced a novel approach for generating crystals with precise symmetry properties. We proposed to leverage asymmetric units and site symmetry representations within a diffusion model framework. This approach ensures that the generated crystals inherently preserve desired symmetries while allowing greater diversity, computational efficiency, and flexibility in the generation process. To encode crystal and site symmetries we introduced a new representation of crystal symmetries that enables information sharing across space groups, improving generalization when learning with a diverse set of crystal symmetries. Our results indicate that this method produces stable, novel, and structurally diverse crystals, with improved computational efficiency, showing promise for discovery in materials science. In this work, we focused on inorganic crystals, but SymmCD could potentially be promising for applications on molecular crystals and co-crystals, which also have non-trivial symmetries. Beyond materials, other data types such as molecules and graphs often exhibit complex symmetries, and future work could investigate if symmetry constraints could also be useful in generative models for these modalities.

A limitation of our framework is that it makes it more challenging to perform crystal structure prediction given a composition since it relies on sampling a space group first, and then a composition conditioned on the space group. Finally, an important area of future work in generative models for crystals is also to go beyond single crystals, and consider the generation of polycrystalline materials. These types of materials are common in applications, yet not suited to generation using single unit cells or asymmetric units.

ACKNOWLEDGMENTS

This project is supported by Intel Labs, CIFAR and the NSERC Discovery grant. S.-O. K.'s research is also supported by IVADO and the DeepMind Scholarships, and D.L.'s research is partly supported by the FRQNT. Computational resources were provided by Mila and Intel Labs. We are thankful to Alexandra Volokhova, Victor Schmidt, Alex Hernandez-Garcia, Alexandre Duval, and Félix Therrien for helpful discussions.

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

## A    REPLICATION

We define the replication operator as $\mathsf{R} : \mathcal{G} \times \mathcal{P} \times \mathbb{R}^3 \to 2^{\mathbb{R}^3}$. This operation is defined by considering the group $S_{\mathbf{x}} \ltimes T_{\mathbf{S}}$, with $T_{\mathbf{S}}$ being the group of translations defined by the lattice. $S_{\mathbf{x}} \ltimes T_{\mathbf{S}}$ is the set of operations that preserve the position of $\mathbf{x}$ *within* the unit cell as opposed to within the crystal. We can then consider the coset decomposition of the space group with respect to that group $G / (S_{\mathbf{x}} \ltimes T_{\mathbf{S}})$. Then, we denote by $[G / (S_{\mathbf{x}} \ltimes T_{\mathbf{S}})]_{\mathbf{0}}$ a system of coset representatives where the translation parts are chosen to move only within the unit cell. This defines the set of operations that move a position $\mathbf{x}$ within its orbit and the unit cell. The replication operation then simply consists of applying all these operations:

$$\mathsf{R}(G, S_{\mathbf{x}}, \mathbf{x}) = ((\mathbf{O}, \mathbf{t})\,\mathbf{x} \mid (\mathbf{O}, \mathbf{t}) \in [G / (S_{\mathbf{x}} \ltimes T_{\mathbf{S}})]_{\mathbf{0}})$$

The representation in terms of individual atoms is then:

$$\mathbf{X} = \bigoplus_i^M \mathsf{R}\left(G, S_{\mathbf{x}'_i}, \mathbf{x}'_i\right) \tag{2}$$

$$\mathbf{A} = \bigoplus_i^M \text{repeat}\left(\mathbf{a}_i, [G : (S_{\mathbf{x}} \ltimes T_{\mathbf{S}})]\right) \tag{3}$$

where $\text{repeat}(\mathbf{a}, n)$ repeats the vector $\mathbf{a}$ $n$ times and $[G : (S_{\mathbf{x}} \ltimes T_{\mathbf{S}})]$ is the multiplicity of the orbit.

In our diffusion model, our predicted site symmetries $\hat{\mathbf{S}}$ do not always necessarily correspond to a valid crystallographic point group. To get around this, we project $\hat{\mathbf{S}}$ to the nearest point group that is a subgroup of the given space group, as measured by the Frobenius Norm of their difference. Once a point group is chosen, the PyXtal `search_closest_wp` function is used to get the nearest coordinates to $\mathbf{X}'$ that correspond to a Wyckoff position with the given site symmetry, and $\mathbf{X}'$ is updated to be placed on those coordinates (Fredericks et al., 2021). Finally, the representative atoms at the Wyckoff position are replicated, using operations implemented in PyXtal.

## B    LATTICE REPRESENTATION

We use the lattice representations derived by Jiao et al. (2024), as they are useful for constraining lattices to respect the symmetries of a given space group. The authors found that any lattice matrix $\mathbf{L}$ can be written as $\mathbf{L} = \mathbf{Q}\exp(\mathbf{S})$ for some orthogonal $\mathbf{Q}$ (which we can ignore, as orthogonal transformations do not change the lattice), and symmetric $\mathbf{S}$. The matrix $\mathbf{S}$ can then be decomposed into a sum of the following basis lattices:

$$\mathbf{B}_1 = \begin{pmatrix} 0 & 1 & 0 \\ 1 & 0 & 0 \\ 0 & 0 & 0 \end{pmatrix}, \quad \mathbf{B}_2 = \begin{pmatrix} 0 & 0 & 1 \\ 0 & 0 & 0 \\ 1 & 0 & 0 \end{pmatrix}, \quad \mathbf{B}_3 = \begin{pmatrix} 0 & 0 & 0 \\ 0 & 0 & 1 \\ 0 & 1 & 0 \end{pmatrix},$$

$$\mathbf{B}_4 = \begin{pmatrix} 1 & 0 & 0 \\ 0 & -1 & 0 \\ 0 & 0 & 0 \end{pmatrix}, \quad \mathbf{B}_5 = \begin{pmatrix} 1 & 0 & 0 \\ 0 & 1 & 0 \\ 0 & 0 & -2 \end{pmatrix}, \quad \mathbf{B}_6 = \begin{pmatrix} 1 & 0 & 0 \\ 0 & 1 & 0 \\ 0 & 0 & 1 \end{pmatrix}.$$

with $\mathbf{S} = \sum_{i=1}^6 k_i \mathbf{B}_i$. They derive constraints on $k_i$ depending on the space groups that a crystal belongs to:

- Triclinic: $\mathbf{k} = (k_1, k_2, k_3, k_4, k_5, k_6)$
- Monoclinic: $\mathbf{k} = (0, k_2, 0, k_4, k_5, k_6)$
- Orthorhombic: $\mathbf{k} = (0, 0, 0, k_4, k_5, k_6)$
- Tetragonal: $\mathbf{k} = (0, 0, 0, 0, k_5, k_6)$
- Hexagonal: $\mathbf{k} = (-\log(3)/4, 0, 0, 0, k_5, k_6)$
- Cubic: $\mathbf{k} = (0, 0, 0, 0, 0, k_6)$

## C  Site Symmetry Representation

The 15 possible symmetry axes of crystals are: [001], [010], [100], [111], [1$\bar{1}$1], [$\bar{1}$11], [$\bar{1}\bar{1}$1], [110], [1$\bar{1}$0], [101], [10$\bar{1}$], [011], [01$\bar{1}$], [210], [120], [1$\bar{1}$0]. These axes are written in short form: for example, [1$\bar{1}$0] denotes the direction of the vector $(1, -1, 0)$. They are shown in Figure 2. The axes depend on the symmetries of the crystal system: for example, in an orthorhombic crystal (a rectangular prism whose side lengths are not necessarily equal), a crystal may have different site symmetries oriented around the x, y, or z-axes. Conversely, in a tetragonal crystal (a rectangular prism with a square base), any site symmetry oriented along the x-axis must also be along the y-axis, there may be additional symmetries along the diagonal of the x-y plane.

The possible set of symmetry elements along each axis for a site symmetry group correspond to the identity 1; an inversion $\bar{1}$; rotations of different orders 2, 3, 4, and 6; rotoinversions $\bar{2}$ (equivalent to a mirror symmetry $m$ across a plane perpendicular to the axis), $\bar{3}$, $\bar{4}$, and $\bar{6}$; and combinations of rotations and mirror reflections $2/m$, $4/m$, and $6/m$. This enumeration yields 13 possible symmetries along each axis.

The possible symmetry elements along each axis for a space group correspond to the identity 1; an inversion $\bar{1}$; rotations of different orders 2, 3, 4, and 6; rotoinversions $\bar{2}$ (equivalent to a mirror symmetry $m$ across a plane perpendicular to the axis), $\bar{3}$, $\bar{4}$, and $\bar{6}$; screws $2_1, 3_1, 3_2, 4_1, 4_2, 4_3, 6_1, 6_2, 6_3, 6_4, 6_5$, and glides $a, b, c, n, d, e$.

To encode a space group, an additional 7-dimensional one-hot encoding is used to denote the Bravais lattice to which the space group belongs. This yields a $(26 \times 15) + 7 = 397$ dimensional binary representation of space group.

The representations can now be accessed using the `symmetry` module of PyXtal (Fredericks et al., 2021).

## D  Diffusion and denoising process details

**Diffusion on lattice parameters k**  Inspired by Jiao et al. (2024), we perform diffusion over **k**, the $O(3)$-invariant lattice representation. The forward noising process is given by $q(\mathbf{k}_t | \mathbf{k}_0) \sim \mathcal{N}(\mathbf{k}_t | \sqrt{\bar{\alpha}_t}\mathbf{k}_0, (1 - \bar{\alpha}_t)\mathbf{I})$, where $\mathbf{k}_t$ is the noised version of $\mathbf{k}_0$ at timestep $t$. Here, similar to Nichol & Dhariwal (2021), $\bar{\alpha}_t = \Pi_{j=1}^t (1 - \beta_j)$, where $\beta_j \in (0, 1)$ determines variance in each step controlled by the cosine scheduler. During the generation process, we start with $\mathbf{k}_T \sim \mathcal{N}(0, \mathbf{I})$ and use learned denoising network to generate $\mathbf{k}_{t-1}$ from $\mathbf{k}_t$:

$$p_\theta(\mathbf{k}_{t-1} | \mathcal{C}_t') = \mathcal{N}\Big(\mathbf{k}_{t-1} | \mu_\mathbf{k}(t), \sigma(t)\mathbf{I}\Big),$$

$$\mu_\mathbf{k}(t) = \frac{1}{\sqrt{\bar{\alpha}_t}}\Big(\mathbf{k}_t - \frac{\beta_t}{\sqrt{1 - \bar{\alpha}_t}}\hat{\epsilon}_\mathbf{k}(\mathcal{C}_t', t)\Big), \sigma(t) = \beta_t \frac{1 - \bar{\alpha}_{t-1}}{1 - \bar{\alpha}_t}.$$

Here, $\mathcal{C}_t'$ is the noised crystal and $\hat{\epsilon}_k(\mathcal{C}_t', t)$ is the predicted denoising term predicted from a denoising network $\phi(\mathcal{C}_t', t)$. We also use a mask $m$ to only implement diffusion over unconstrained dimensions of $\mathbf{k}_t$, since depending upon space groups, certain dimensions have fixed values (Appendix B). The mask can be represented as $m \in \{0, 1\}^6$ and $m_i = 1$ indicates that $i^{th}$ index of $\mathbf{k}$ is unconstrained. The corresponding loss used to train the denoising network is:

$$\mathcal{L}_\mathbf{k} = \mathbb{E}_{\epsilon_\mathbf{k} \sim \mathcal{N}(0, \mathbf{I}), t \sim \mathrm{U}(1, T)}[||m \odot \epsilon_\mathbf{k} - \hat{\epsilon}_\mathbf{k}(\mathcal{C}_t', t)||_2^2]$$

where $\odot$ is the elementwise product and $U(1, T)$ is a uniform distribution over timesteps.

**Diffusion over representative fractional coordinates $\mathbf{X}'$**  We perform diffusion over the fractional coordinates using the same method as Jiao et al. (2023). Due to the periodicity of fractional coordinates, the noising process $q(\mathbf{X}_t | \mathbf{X}_0)$ is determined by a Wrapped Normal distribution rather than a Gaussian distribution, and we initialize the fractional coordinates $\mathbf{X}_T$ with the uniform distribution $U(0, 1)$ when sampling.

**Diffusion on atom types $\mathbf{A}'$** We use discrete diffusion from Austin et al. (2021) to sample the atom types of each representative. If $\mathbf{a}_0 \in \{0, 1\}^Z$ is the one-hot encoding of atom types for a single representative, then we can noise it as: $q(\mathbf{a}_t | \mathbf{a}_0) = \mathrm{Cat}(\mathbf{a}_t; \mathbf{p} = \mathbf{a}_0^\top \bar{\mathbf{Q}}_t)$, where $\bar{\mathbf{Q}}_t = \prod_{i=1}^t \mathbf{Q}_i \in \mathbb{R}^{Z \times Z}$ is the cumulative product of transition matrices between timesteps. Inspired by Vignac et al. (2023), the transition matrix can be parametrized as $\mathbf{Q}_t = \alpha_t \mathbf{I} + \beta_t \mathbf{m}_a$, where $\mathbf{m}_a$ are the marginals over the atom types in the data, and $\alpha_t$ and $\beta_t$ are scheduling parameters. The effect of this noising scheme is that regardless of $\mathbf{a}_0$, the fully noised $\mathbf{a}_T = \mathbf{a}_0^\top \mathbf{Q}_T = \mathbf{m}_a$, so we can sample from the prior distribution $\mathbf{m}_a$, which is close to the data distribution. The discrete diffusion model is trained using a cross-entropy loss:

$$\mathcal{L}_{\mathbf{A}'} = \mathbb{E}_{\mathbf{a}_t \sim \mathrm{Cat}(\mathbf{a}_0^\top \bar{\mathbf{Q}}_t), t \sim \mathrm{U}(1,T)} \sum_{i=1}^M \mathrm{CrossEntropy}(\mathbf{a}_i', \hat{\mathbf{a}}_i'), \tag{4}$$

where $\hat{\mathbf{a}}_i$ are the probabilities predicted by the model $\phi(\mathcal{C}_t', t)$. To sample from the discrete diffusion model, we sample from the marginal distribution over atom types $\mathbf{m}_a$, then progressively denoise using:

$$q(\mathbf{a}_{t-1} | \mathbf{a}_t, \mathbf{a}_0) = \mathrm{Cat}\left(\mathbf{a}_{t-1}; \mathbf{p} = \frac{\mathbf{a}_t^\top \mathbf{Q}_t^\top \odot \mathbf{a}_0^\top \bar{\mathbf{Q}}_{t-1}}{\mathbf{a}_0^\top \bar{\mathbf{Q}}_t \mathbf{a}_t}\right) \tag{5}$$

More details of this implementation can be seen in Vignac et al. (2023).

**Diffusion for site symmetries $\mathbf{S}$** The site symmetry representation matrices described in Section 4.2 can be thought of as 15 separate 13-dimensional categorical variables: one site symmetry operation per axis. Our diffusion model over site symmetries is almost identical to the method for atom types, applying discrete diffusion separately over each of the axes. Because the site symmetries depend strongly on the space group, we use transition matrices that are different for each space group: $\mathbf{Q}_{t,i,G} = \alpha_t \mathbf{I} + \beta_t \mathbf{m}_{S_\alpha, G}$, where $\mathbf{m}_{S_u, G}$ denotes the marginals over site symmetry operations for axis $S_u$ given space group $G$. For each representative node, we average the cross-entropy loss over each of the axes.

# E  ARCHITECTURE DETAILS

## E.1  DENOISING MODEL

We use a graph neural network based on the architecture of Jiao et al. (2023). We embed the timestep $t$ using sinusoidal embeddings, $\psi_t(t)$. We embed our space group representation from Section 4.2 using an MLP, $\phi_G(G)$. We embed our site symmetries by separately embedding each axis using the same network, and feeding the resulting embeddings into a secondary MLP: $\phi_S(\bigoplus_{u=1}^{15} \phi_U(S_u))$. These are all used to initialize the node embeddings $\mathbf{h}_i$.

$$\mathbf{h}_i \leftarrow \phi_h\left(\mathbf{a}_i, \mathbf{x}_i, \phi_S\left(\bigoplus_{u=1}^{15} \phi_U(S_u)\right), \phi_G(G), \psi_t(t)\right).$$

As noted earlier, we directly use coordinates $\mathbf{x}$, because we are working a conventional or canonical lattice, and so Euclidean symmetries are not necessarily useful here.

At each layer we compute messages and use them to update node embeddings:

$$\mathbf{m}_{ij} \leftarrow \phi_m(\mathbf{h}_i, \mathbf{h}_j, \mathbf{k}, \psi(\mathbf{x}_i - \mathbf{x}_j))$$

$$\mathbf{h}_i \leftarrow \mathbf{h}_i + \phi_h(\mathbf{h}_i, \sum_j^M \mathbf{m}_{ij})$$

Here, $\psi$ is a Fourier embedding, $\phi_m$ and $\phi_h$ are MLPs acting on edges and nodes respectively. We use a SiLU activation function for each MLP. Finally, we output predicted $\hat{\epsilon}_{\mathbf{X}'}$, $\hat{\mathbf{A}}'$ and $\hat{\mathbf{S}}$ using the node embeddings $\mathbf{h}_i$, and $\hat{\epsilon}_{\mathbf{k}}$ using $\sum_i^M \mathbf{h}_i$.

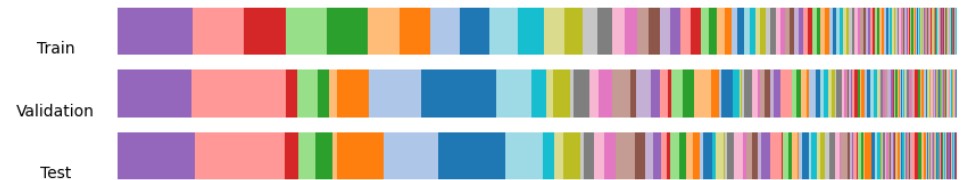

Figure 5: Proportion of space group symmetries of the MPTS-52 dataset. From left to right, the first few space groups are: Pnma, P2₁/c, Fm$\bar{3}$m, I4/mmm, P6₃/mmc, Pm$\bar{3}$m, C2/m, C2/c, $\bar{1}$, and Cmcm.

Table 5: Template statistics for MPTS-52

| Method | # Unique | % in Train | # New |
|---|---|---|---|
| Training Set | 4452 | 100 | - |
| SymmCD | 2772 | 48.3 | 1432 |

Table 6: Percent of stable and S.U.N. samples produced from an initial set of 10,000 generated crystals for SymmCD trained on the MPTS-52 dataset.

| Initial Stable | Relaxed Stable | Relaxed S.U.N. |
|---|---|---|
| 1.72% | 5.97% | 4.62% |

Table 7: The validity, coverage, and property distribution metrics for SymmCD trained on the MPTS-52 dataset.

| Validity (%) (↑) | | Coverage (%) (↑) | | Property Distribution (↓) | | | |
|---|---|---|---|---|---|---|---|
| Struct. | Comp. | Recall | Precision | $d_\rho$ | $d_E$ | $d_{\text{elem}}$ | $d_{\text{sg}}$ |
| 90.1 | 79.2 | 99.6 | 96.3 | 0.844 | 0.549 | 0.317 | 0.274 |

### E.2 MODEL HYPERPARAMETERS

The graph neural network has 8 layers, and we use a representation dimension of 1024 for $\mathbf{h}_i$. We encode distances between nodes using a sinusoidal embedding, with 128 different frequencies. We encode the timestep $t$ into a 10 dimensional vector. We apply layer normalization at each layer of the GNN. The loss coefficients selected were $\lambda_{\mathbf{k}} = 5$, $\lambda_{\mathbf{X}'} = 1$, $\lambda_{\mathbf{A}'} = 0.1$ and $\lambda_{\mathbf{S}} = 10$.

We performed two hyperparameter sweeps: we first tested each combination of $\lambda_{\mathbf{k}}$, $\lambda_{\mathbf{A}'}$ and $\lambda_{\mathbf{S}}$ set to values in $\{0.1, 0.5, 1, 5, 10\}$, while keeping $\lambda_{\mathbf{X}'}$ fixed at 1, and then selected the loss coefficients that lead to the highest structural validity. Next, we performed a random sweep of other architecture parameters, running 150 different hyperparameter combinations and choosing a model that had high performance on structural validity, compositional validity, and $d_E$. We varied the number of GNN layers in $\{6, 8, 12, 16\}$, representation dimension in $\{256, 512, 1024\}$, time embedding dimension in $\{10, 64, 256\}$, and varied whether layer normalization was used.

## F EXTENDED RESULTS

### F.1 MPTS 52 DATASET

In addition to MP-20, we also trained SymmCD on the MPTS-52 dataset (Baird et al., 2024), a more challenging subset of the Materials Project that contains materials with up to 52 atoms per primitive unit cell. Unlike MP-20, it does not filter out materials containing radioactive elements. The dataset contains 40,476 samples with a train/validation/test split of 27,380/5,000/8,096 crystals. The splits are in chronological order, with the materials in the test set having been discovered most recently, and the materials in the training set having been discovered earliest. The distribution of space groups found in the dataset are shown in Figure 5. None of the diffusion and flow-based methods we compared against (CDVAE, DiffCSP, DiffCSP++, or FlowMM) have reported de-novo generation results on this dataset.

We trained SymmCD on MPTS-52 using all of the same hyperparameters as were used for the MP-20 dataset, but trained for 1500 epochs. We sampled 10,000 crystals, and checked the same proxy metrics. We also relaxed the generated crystals using CHGNet and checked for whether the generated crystals were S.U.N. The unique templates (as described in Section 5.1) of the dataset and of our generated crystals are shown in Table 5. The results for the proxy metrics are shown in Table 7 and stability results are shown in Table 6.

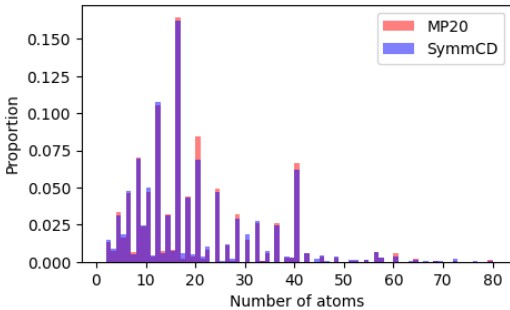

Figure 6: Histogram of the number of atoms in crystals from MP-20 and generated by SymmCD when trained on MP-20.

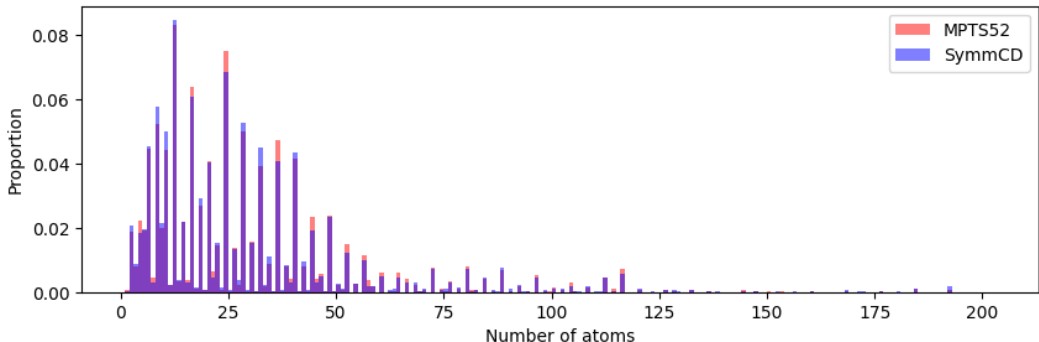

Figure 7: Histogram of the number of atoms in crystals from MPTS-52 and generated by SymmCD when trained on MPTS-52.

## F.2 NUMBER OF ATOMS

To demonstrate that SymmCD is able to correctly predict reasonable site symmetries, we show here that the distribution of number of atoms per crystal matches the dataset it is trained on. This is not a trivial task, as the model needs to learn the multiplicity of different possible site symmetries, which depends on both the different symmetry elements of the site symmetry and the space group that it belongs to. The comparison for MP-20 is shown in Figure 6, and the comparison for MPTS-52 is shown in Figure 7.

## F.3 PROPERTY PREDICTION TASK

We test the usefulness of our site symmetry representation using a regression experiment. We selected formation energy per atom as the target property to predict.

We use DimeNet++ (Gasteiger et al., 2020b;a) as a base model to perform ablation over the type of input graph and encoding site symmetry information per node.

One input format is a multi-graph (Xie et al., 2022), which describes the unit cell as a graph with nodes as atoms and edges between them according to a cutoff radius. These edges could potentially span to neighbouring unit cells. The other input format is the asymmetric unit that we use in SymmCD. Under these two inputs, we test the effects of including a site symmetry encoding for each node. We report the Mean Absolute Error (MAE) for the test set in Table 8. We see that the effect of including site symmetry

Table 8: Mean average error when predicting crystal formation energy. The input could be the asymmetric unit or a multi-graph, and the site symmetry information can be encoded or ignored. We observe that our encoding of site symmetry helps predict the target property.

|  | Multigraph | Asymm. Unit |
|---|---|---|
| W/out $\mathbf{S}$ | 0.0214 | 0.0711 |
| With $\mathbf{S}$ | 0.0212 | 0.0490 |

information is minimal when we have access to the full
graph. However, we see that when we are restricted to only using the asymmetric unit, having access
to the site symmetry info greatly helps, showing that we can recover some geometric information
lost when using just an asymmetric unit by also including symmetry.

## F.4 EXAMPLES

In Figure 8, we include 6 randomly sampled crystals generated by SymmCD along with their respective space groups.

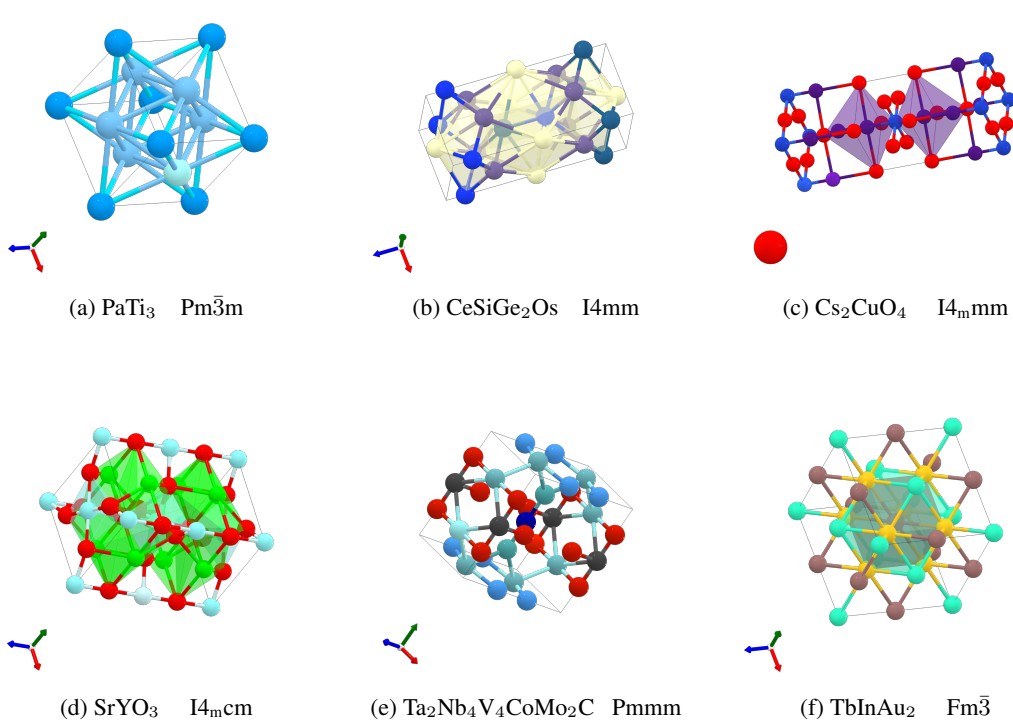

(a) $PaTi_3$    $Pm\bar{3}m$       (b) $CeSiGe_2Os$    I4mm       (c) $Cs_2CuO_4$    $I4_mmm$

(d) $SrYO_3$    $I4_mcm$       (e) $Ta_2Nb_4V_4CoMo_2C$  Pmmm       (f) $TbInAu_2$    $Fm\bar{3}$

Figure 8: Example materials generated by SymmCD, along with their chemical formulate and space groups symmetries.

## F.5 PROXY METRICS

We compare the different methods using the metrics established by Xie et al. (2022), measuring
the validity, coverage, and property statistics of the generated crystals. We measure the validity by
checking structural validity, defined as whether no two atoms are closer than 0.5 Å apart, and com-
positional validity, defined as whether the charges are balanced as determined by SMACT (Davies
et al., 2019). It should be noted that the compositional validity of the MP-20 dataset is only 92%.
To determine coverage, we examine the CrystalNN structural fingerprints (Zimmermann & Jain,
2020) and Magpie compositional fingerprints (Ward et al., 2016) of the valid generated crystals,
and look at their distances to the fingerprints of the crystals in the test set. If the distance is under
some cutoff, then the crystals are matched, giving both recall and precision metrics. We look at the
distances between the properties of the valid generated crystals and the crystals from the test set
to compare the ability of each model to match the data distribution. We specifically compare the
Wasserstein distances between the atomic densities $d_\rho$, number of unique elements $d_{elem}$, and the
formation energy $d_E$ predicted by a pretrained DimeNet++ model (Gasteiger et al., 2020a). We also
look at the Jensen-Shannon distance between the space groups of the generated valid structures and
the space groups of the crystals in the test set, denoted by $d_{sg}$. The coverage and property statistics
are computed only for a random subset of 1000 valid crystals per method. For each method, we

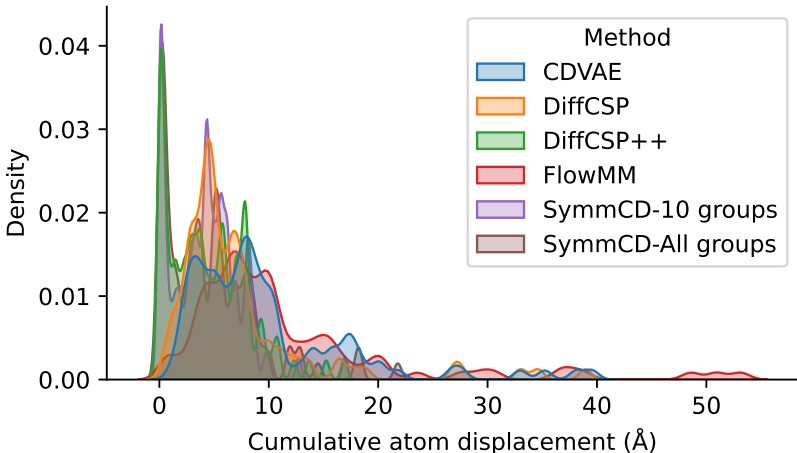

Figure 9: Distribution of total atom displacements over the course of relaxation. Displacements are averaged over structures for a given method.

train 5 different models with different random seeds so that we could see the variance for each of the metrics.

## F.6 DENSITY FUNCTIONAL THEORY

We performed cell and geometry relaxation calculations using the PBE functional, `DZVP-MOLOPT-SR-GTH` basis set, and `GTH-PBE` pseudopotential with the `QUICKSTEP` program from `CP2K` (Perdew et al., 1996; Kühne et al., 2020). Table 9 shows the settings—particularly convergence thresholds—used for performing the relaxations. Values were generally tuned to balance between a feasible computational budget and a fair benchmark between methods: as an example, we allow relaxations to attempt to continue even with poorly convergent SCF in the hopes that optimization trajectories will still have a chance of finding a local minimum. With that in mind, we note that the convergence thresholds are also generally set to relatively "lax" values, and for end property values stricter convergence thresholds may be necessary.

Figure 9 shows distributions of the cumulative atom displacement, averaged over structures for each method: in other words, the total distance the average atom travels over the full course of the relaxation. Naturally, the less atom displacement, the more likely a given generative method produces high fidelity crystal structure samples. We see that DiffCSP++ and both treatments of SymmCD produce samples with displacements that peak close to zero, while other methods peak closer to ~4.5Å. The relatively long tails out to high cumulative displacements seen with CDVAE and FlowMM are attributed to trajectories that do not converge after the specified number of relaxation steps. From this, we can conclude that with the exception of SymmCD and DiffCSP++, the tested generative methods demonstrate a similar degree of fidelity requiring some degree of optimization.

Figure 10 shows empirical distribution functions of the maximum gradient value at the end of the relaxation trajectory, regardless of the state of convergence: the smaller the maximum gradient is, the closer the relaxation ended in a local minimum. The first observation is that across all methods, a significant portion (~60%) of trajectories fail to converge (the portion to the right of the dashed line)—the majority of sampled structures fail to converge, based on our naive attempt to relax them using our reasonable choice of method, basis set, and pseudopotential. Another observation is the long tail towards low values for the 10 space group treatment of SymmCD, which provides compelling evidence for extremely high fidelity samples being produced by SymmCD. The expected values of the maximum gradient for each method are shown in Table 10, where we can observe that SymmCD tends to have a lower maximum gradient after relaxation.

Table 9: Configuration settings for CP2K. Settings that are omitted from this table assume their default values.

| Parameter | Value |
|---|---|
| **Base SCF settings** | |
| `EPS_SCF` | $10^{-7}$ |
| `MAX_SCF` | 300 |
| `MAX_ITER_LUMO` | 400 |
| `IGNORE_CONVERGENCE_FAILURE` | T |
| **Orbital transformation** | |
| Orbital transformation method | `IRAC` |
| `ENERGY_GAP` | $10^{-3}$ |
| `MINIMIZER` | `DIIS` |
| `LINESEARCH` | `2PNT` |
| `PRECONDITIONER` | `FULL_ALL` |
| **Outer SCF settings** | |
| `MAX_SCF` | 20 |
| `EPS_SCF` | $10^{-6}$ |
| **Cell optimization** | |
| `TYPE` | `DIRECT_CELL_OPT` |
| `MAX_ITER` | 100 |
| `OPTIMIZER` | `BFGS` |
| **Geometry optimization** | |
| `MAX_DR` | $3 \times 10^{-3}$ |
| `MAX_FORCE` | $9 \times 10^{-4}$ |
| `RMS_DR` | $1.5 \times 10^{-3}$ |
| `RMS_FORCE` | $6 \times 10^{-4}$ |
| `MAX_ITER` | 100 |
| `OPTIMIZER` | `BFGS` |
| **BFGS** `TRUST_RADIUS` | 0.25 |

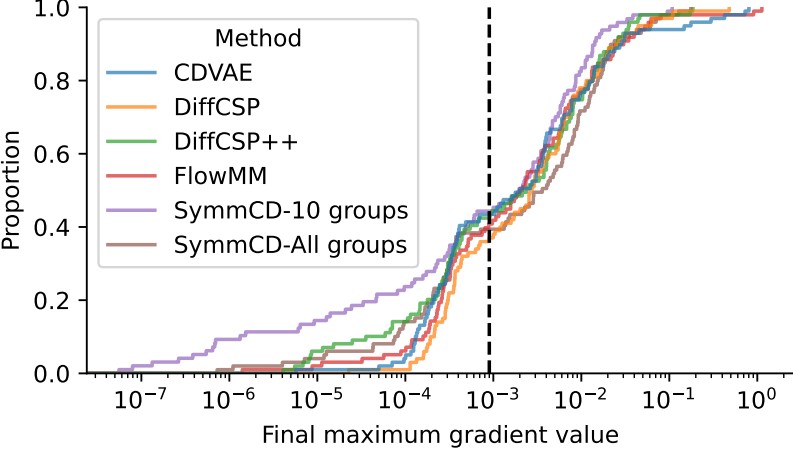

Figure 10: Empirical cumulative distribution plots for the maximum, absolute value of the gradient across atoms after relaxation, regardless of convergence. The dashed line indicates the convergence threshold stated in Table 9.

Table 10: Expected values of the maximum gradient for each method from integrating the curves in Figure 10.

| Method | $\mathbb{E}[|\nabla|_{\max}]$ ($\downarrow$) |
|---|---|
| DiffCSP | 0.000494 |
| CDVAE | 0.000235 |
| FlowMM | 0.000083 |
| DiffCSP++ | 0.000072 |
| SymmCD (All groups) | 0.000057 |
| SymmCD (10 SGs) | 0.000038 |

