# OpenReview forum: "SymmCD: Symmetry-Preserving Crystal Generation with Diffusion Models"
_ICLR.cc/2025/Conference — ICLR 2025 Poster_

### Official Review · Reviewer_B8yV · 2024-11-02

**Soundness:** 3
**Presentation:** 3
**Contribution:** 3
**Rating:** 8
**Confidence:** 3

**Summary:**

This paper presents a novel diffusion-based generative model, SymmCD, for symmetry-preserving crystal generation. The proposed approach explicitly incorporates crystallographic symmetry into the generative process, using a unique representation that decomposes crystals into asymmetric units and symmetry transformations. This design enhances both computational efficiency and the diversity of generated crystal structures, addressing some limitations of existing models in terms of symmetry and structural validity.Overall, the paper presents a strong contribution to the field of crystal generation. By explicitly incorporating symmetry into a generative diffusion framework, SymmCD addresses critical limitations of prior methods and provides a promising tool for materials discovery.

**Strengths:**

1. Innovation in Representation: The paper introduces a physically motivated representation based on crystallographic symmetry, using a binary matrix to encode symmetry, which addresses data fragmentation and enables generalization across symmetry groups.
2. Computational Efficiency: By focusing on asymmetric units rather than full crystal structures, the model demonstrates significant improvements in memory usage and training speed, an aspect well-supported by experimental evidence.
3. Diversity and Validity of Generated Structures: SymmCD shows impressive results in generating diverse, valid, and symmetry-conforming crystal structures across multiple symmetry groups, even those that are less common in training data.

**Weaknesses:**

1.Comprehensive Evaluation of Generated Crystal Properties: While the model’s ability to generate symmetric and diverse crystals is demonstrated, additional quantitative evaluations of properties such as thermodynamic and mechanical stability would further solidify the model’s applicability to real-world scenarios. Metrics that reflect physical applicability, such as structural stability under various conditions, could significantly strengthen the evaluation section.
2.Efficiency on Larger Datasets: SymmCD’s efficient crystal representation is highlighted as a key advantage. However, a more comprehensive analysis of its computational efficiency on larger datasets, or under different hardware setups, could provide a more complete understanding of its scalability and practical utility in materials science applications.
3.Clarification of the Binary Symmetry Encoding: The binary matrix representation for symmetry is an intriguing solution to data fragmentation, yet further explanation on why this approach outperforms traditional encodings in practical settings would be beneficial. Additional details in the architecture and experimental sections could clarify how the representation is effectively utilized in training.
4.It may be helpful to provide a clearer explanation of the training algorithm, particularly in how the diffusion and denoising processes maintain symmetry.

**Questions:**

Since I am not a researcher in this field, I don’t know much about the specific background, so I am not very clear about the process shown in Figure 3. Figure 3 and the training pipeline section could benefit from additional annotations to improve readability for those unfamiliar with diffusion models in this context.

---

> ### Author Response · Authors · 2024-11-19
> **Response to Reviewer B8yV**
>
> Thank you for your thoughtful comments. We are glad the reviewer found that our paper presents an innovative method, with impressive performance and improved computational efficiency compared to previous approaches.
>
> Below, we  address the reviewer’s comments:
>
> - **Thermodynamic and mechanical stability evaluation**: This is an interesting point! The issue is that evaluation of thermodynamic or mechanical stability would require very expensive DFT simulations that cannot realistically be run on hundreds of samples for evaluation.
> - **Efficiency on larger datasets:** We agree that our method has potential to scale to larger datasets more efficiently than other methods. To the best of our knowledge, the only larger publicly available dataset of inorganic crystal structures is the MPTS-52 dataset. No other generative modeling method has been able to scale to this dataset. As a direction for future work, we are investigating if our model could be trained on this dataset. This would be a significant breakthrough, but requires more computational resources than what we currently have access to.
> - **Clarification of binary symmetry encoding**: We have clarified this in the updated manuscript to take your feedback into account. We have added statistics to show how data fragmentation is an issue for methods that do not use a shared representation over symmetries. Specifically, for the MP-20 dataset it turns out that 113 space groups out of 169 in the training set have fewer than a hundred samples associated with them, which is a very small amount of data.
> - **Generation process preserving symmetry and clarification of figure 3**: This is a useful comment! We have added a short explanation as to why the diffusion and denoising process symmetry. Essentially, it is because they only operate on the asymmetric unit, which is the minimal representation given a symmetry. In an effort to improve clarity, we have updated figure 3 with additional annotations as you suggested. The goal of the updates is to better highlight some elements specific to the training and sampling from the diffusion model that might have been confusing in the previous version.
>
> We thank you again for your review. Note that all the changes in the revised manuscript have been highlighted in blue for clarity. Please let us know if we have addressed your concerns appropriately or if we can answer any other question.

---

### Official Review · Reviewer_Tsh1 · 2024-11-03

**Soundness:** 3
**Presentation:** 3
**Contribution:** 4
**Rating:** 8
**Confidence:** 4

**Summary:**

The authors present the generative model SymmCD, which allows for the generation of datasets of crystalline structures of non-molecular crystals while explicitly considering symmetry. The results obtained exhibit both high symmetry diversity and a significant percentage of thermodynamically stable structures, making SymmCD a solid choice for crystal structure prediction systems or virtual screening of crystalline materials.

**Strengths:**

- The developed method for vectorizing crystalline structures, which explicitly accounts for both the spatial symmetry of the crystal and the point symmetry of the orbits, is to my knowledge the first of its kind, therefore unique, and holds a great promise for application in crystal structure prediction (CSP) for both inorganic and organic crystals.
- The article is well-structured and clearly conveys information, allowing individuals unfamiliar with this field to understand the crystallographic features of the problem with some investment of time.
- I believe this work could be highlighted at the conference as a fine example of how rational design of vector representation can influence the overall effectiveness of the developed deep learning model.

**Weaknesses:**

The major weaknesses of the paper lie in the discussion of the obtained empirical results. Addressing these will significantly enhance the presentation of the work accomplished:

1. Please indicate in the introduction that the initial focus is on non-molecular/inorganic crystals.

2. Please mention in the conclusion remarks that your method of structural representation seems to be well-suited for molecular crystals as well. For the latter, the presence of intrinsic point symmetry and its interaction with the point symmetry of orbitals is one of the key factors determining the crystal structure.

3. In your conclusions, when you state "go beyond single crystals, and consider generating multi-component crystals and alloys," please clarify what you mean. "Single crystal" is a broad term contrasting with polycrystalline materials and does not directly relate to crystalline structure. A multi-component crystal refers to a crystal composed of multiple chemical substances; for instance, this includes pharmaceutical co-crystals. Clearly, your approach should be applicable to these systems.

4. Please consider rewriting conclusions to emphasize advantages (applicability to molecular crystals, including co-crystals) rather than deficiencies (inapplicability to non-crystalline systems), but also, to provide a deeper discussion of the limitations of SymmCD along with practical implications for the actual industrial problems.

**Questions:**

1. In Table 2, why is CDVAE bold instead of SymmCD (10 SGs) for Validity Comp.? I understand an argument could be made that comparing the 10 SGs version to the other methods might not be entirely appropriate, but then I would consider dropping this version from tables 1 and 2, and only discuss it in the context of table 3, where the S.U.N. shines. Please clarify the logic behind inclusion of SymmCD (10 SGs) for Table 1 and Table 2.
2. The evaluation presented in Table 3 involves random subsampling of 10% of the generated crystals, followed by two predictive models to evaluate stability and S.U.N. properties. At the same time, the SymmCD shows only a marginal improvement compared to DiffCSP and DiffCSP++. Are these results statistically significant? Please provide details on the robustness of this evaluation.
3. Which model appears in the Table 4 as Conventional Unit Cell? Please provide a citation and clarify how this model was included in other comparisons as well.
4. Is it possible to provide a link to the anonymized repository reproducing experiments?

---

> ### Author Response · Authors · 2024-11-19
> **Response to Reviewer Tsh1**
>
> Thank you for your comments on our paper and your supportive review. We are happy to see your positive assessment of the writing of the paper and the feedback regarding the usefulness of our crystal symmetry and more broadly crystal structure representation.
>
> Below, we address some of your comments more concretely:
>
> - **1-4 (applications on inorganic and organic crystal)**: This is a useful comment. We have updated our draft to clarify that our main focus relates to inorganic, solid-state materials. Given the crucial importance of symmetry for inorganic materials, we thought this would be a useful start. As you correctly point out, symmetry-informed representations can be useful for a wider range of applications including molecular crystals. While many parts of our method would be directly applicable to those cases, we believe future work is needed to fully develop and validate such a method. The reviewer is correct in pointing out that our method can be applied to multi-component crystals. We have corrected this, we meant to point out polycrystalline materials as an avenue of future work since they cannot be described by a single unit cell. We have also reframed the conclusion to emphasize the advantages of SymmCD related to generation and focus on the limitations and some of the important practicalities beyond generative methods, such as materials synthesis of complex materials.
> - **Q1 (10 SGs variant)**: Great suggestion! We initially included the 10SGs version for the sake of being complete. However, the goal with this variant of the model is to show that sampling from the most frequent space groups will result in overall more stable materials, as shown in the SUN metrics in table 3 and structural validity in table 2. Since it is pretty obvious that the effect of conditioning on specific space groups will be to reduce diversity and alignment with the dataset, we have applied the reviewers suggestion of removing this variant from tables 1.
> - **Q2 (evaluation)**: We agree that the margins of difference are not large enough to make strong claims about the superiority of SymmCD over other methods when it comes to evaluating stability.
> Unfortunately it is quite slow to perform structural relaxations. But, in the light of your feedback, we are currently working on performing relaxations on the full 10,000 samples and will update the paper when they are completed.
> - **Q3 (conventional unit cell model)**: The comparison is between our proposed SymmCD and DiffCSP with the same architecture and hyperparameters. Note that SymmCD leverages the asymmetric unit and learns the site symmetries while DiffCSP uses the entire conventional unit cell which leads to poor computational efficiency metrics. We previously highlighted this in Section 5.4. However, in the revised version, we have also cited DiffCSP.
> - **Q4 (anonymized code)**: Yes! We have uploaded an anonymized version of the repository here: https://anonymous.4open.science/r/SymmCD-596C/. We have also included a link in the paper.
>
> We thank you again for your review. Note that all the changes in the revised manuscript have been highlighted in blue for clarity. Please let us know if we have addressed your concerns appropriately or if we can answer any other question.

---

> > ### Comment · Reviewer_Tsh1 · 2024-11-24
> >
> > Thanks to the authors for the responses and new additions to the manuscript!
> > I have no further questions and my score remains the same.

---

### Official Review · Reviewer_7azh · 2024-11-03

**Soundness:** 3
**Presentation:** 4
**Contribution:** 2
**Rating:** 5
**Confidence:** 3

**Summary:**

The submission, "Symmetry-Preserving Crystal Generation with Diffusion Models," proposes a method for generating single-crystal structures with precise symmetric properties. The authors use asymmetric units and site symmetry representation, followed by a diffusion model for generation. This method explicitly addresses the generation of crystals with respect to their symmetry group.
The method performs on par with existing approaches but has a lower computational footprint.

**Strengths:**

•	The manuscript's structure and clarity are excellent overall.

•	The manuscript includes a well-written and comprehensive introduction, with a clear and well-developed motivation for the crystal generation problem as an application of diffusion models.

•	The method is well-formalized and understandable even to non-experts in crystal generation.

•	Experimental tasks and evaluation: The authors assess their method and the baselines on relevant additional tasks, such as S.U.N. structure prediction and other proxy metrics, which highlight the proposed method's strengths.

**Weaknesses:**

•	Introduction: From my perspective, the problem of generating symmetric crystals is closely related to other structure generation tasks in general representation learning. For instance, in biological applications, such as neuron structure generation or vascular structure generation, it would be beneficial if the authors discussed the relation to other domains in structure generation and the types of methods that have been developed. For example, I see certain similarities to diffusion methods in molecule generation [https://ieeexplore.ieee.org/abstract/document/10419041] or graph generation [https://arxiv.org/abs/2209.14734], which are partly mentioned in the methods since they are used; however, a discussion of how these applications relate to the context of representation learning would be valuable.

•	Reproducibility: I did not find a link to an anonymous repository or source code in OpenReview, hindering the evaluation of reproducibility for this submission.


•	Experimentation: There are only minor performance gains (if any) compared to the state of the art. What are the practical uses of crystal symmetry generation in academia or industry? Is the computational gain truly relevant, considering the regular applications and scenarios in which crystal symmetry generation methods are used?

•	Experimentation: "We withhold 20% of the dataset as a validation set, and 20% as a test set" (Line 377). The experimental setup suggests that the authors do not use a form of cross-validation or cross-testing. Is there a specific reason for this choice? Given that the authors describe their computational efficiency as a strength, extensive cross-validation across experiments would seem reasonable.

•	Experimentation: Hyperparameter Selection (Section E.2). The authors briefly describe their final hyperparameters: “These hyperparameters were chosen using a sweep” (Line 919). Without code availability and the validation issues mentioned earlier, this appears to be a limited experimental description. What was the hyperparameter search space/budget? How were the hyperparameters for the four baselines tuned exactly? The results show very small differences in performance, so a fair description of hyperparameter search is crucial for reproducibility.


I am not an expert in crystal generation and potentially some of my questions are atypical in the field, I am curious to hear the authors and other reviewer comments and willing to change my rating accordingly.





## Post rebuttal and discussion period comment:

I want to thank the authors for replying and interacting in the discussion period. While the authors reply to all comments, I am not fully convinced by their replies.

I am still convinced that the experimentation is not rigorous and sufficiently validated. Unfortunately, the authors do not reply to the very concrete questions I asked; see details in an extra comment.

While I still like the idea and methodological contribution, I am worried about reproducibility, and I am slightly lowering my score and kindly ask the ACs to discuss the reproducibility aspect when considering the paper for acceptance.

**Questions:**

Please consider the questions raised in the Weaknesses section.

---

> ### Author Response · Authors · 2024-11-19
> **Response to Reviewer 7azh**
>
> Thank you for your comments on our paper. We appreciate your positive feedback on the clarity of our writing, given the complexity of the topic, and on the soundness of the method. Below, we address some of your comments more concretely:
>
> - **Introduction**: We appreciate your suggestion of tying in our work to a broader context of structure generation. We see our work as a continuation of a more general line of research on generative modeling for science, but applied to the domain of materials design. We focused our approach initially on inorganic materials given the crucial importance of symmetry for that class of materials. You correctly point out that symmetry is relevant for a broader diversity of systems, including molecules (organic materials) and graph generation which often use the discrete diffusion framework. SymmCD’s approach leveraging an asymmetric unit is indeed more applicable to molecules and molecular crystals where such building blocks are commonly available. We have updated the related works section to make this connection clearer. We also agree that there are useful intersections between our method and other design cases from a representation learning perspective and have updated the conclusion to reflect that. We believe that conditioning on symmetry can lead to powerful representations that can be applied to generative methods for a wide range of use cases.
> - **Reproducibility**: This is a valid point! We have uploaded an anonymized version of the repository here to address this: https://anonymous.4open.science/r/SymmCD-596C/. We have also included a link in the paper.
> - **Experimentation (performance gain)**: While SymmCD performs on par with other methods on metrics such as stability, validity, and distributions of properties, our main claim is that SymmCD performs significantly better than prior works at generating crystals with *realistic, diverse symmetries*, as seen in Figure 4 and Table 1. The inability of prior works to address this issue has been noted by others [1] and represents a significant barrier to useful crystal generation, as the vast majority of real-world crystals have complex symmetries. Many properties of crystals (such as piezoelectricity and optical activity) are determined by symmetry, so when searching for practical crystals, a generative model should be able to generate crystals with desired symmetry. The computational speedup is also important in practical settings, as high-throughput materials discovery requires the generation of tens of thousands of crystals.
> - **Experimentation (validation)**: We use the same standardized train/validation/test set split as the methods we compare against, for fair comparison. Unfortunately, it would have been too costly to run cross-validation for each baseline method, and this is not standard practice.
> - **Experimentation (hyperparameters)**: We performed two hyperparameter sweeps: we first checked all values of $\lambda_k$, and $\lambda_S$, keeping $\lambda_X$ fixed at 1, and selected the loss coefficients that lead to the highest structural validity. Next, we performed a random sweep of other architecture parameters, running 150 different hyperparameter combinations and choosing a model that had high performance on  structural validity, compositional validity, and $d_E$. We’ve added details in appendix E.2, and included the hyperparameter sweep configuration file in the repository. For all baselines, we use their reported hyperparameters.
>
> We thank you again for your useful feedback. Note that all the changes in the revised manuscript have been highlighted in blue for clarity. Please let us know if we have addressed your concerns appropriately or if we can answer any other questions.
>
> [1] Anthony K Cheetham and Ram Seshadri. Artificial intelligence driving materials discovery? perspective on the article: Scaling deep learning for materials discovery. Chemistry of Materials, 36 (8):3490–3495, 2024.

---

> ### Comment · Reviewer_7azh · 2024-11-22
> **Additional post rebuttal reviewer comments**
>
> Dear authors and fellow reviewers,
>
> I appreciate your rebuttal comments to my and other reviews. Especially, providing the code is an important transparent step.
>
> However, after reading the review of Reviewer Tsh1 and your reply I am now even more concerned about the experimentation, reproducibility and robustness of your presented results.
>
> I have a few follow up questions and concerns and would like to hear your and other reviewer comments.
>
> 1. Hyperparameter selection. I am very concerned about the fact that the authors do not carry out any hyperparameter search for the baseline but do a hyperparameter search for their own method. I have compared the baseline performances reported in the DiffCSP++ paper by Jie et al. and the FlowMM paper by Miller et al. to the ones the authors reported in Table 2 of this manuscript and find quite a difference in Property distribution metrics reported. The metrics of the baselines here appear to be mostly lower. I cannot evaluate how these metrics relate to the template statistics reported in Table 1 but since there is no crossvalidation of the presented work and hyperparameter search for the baselines I am concerned that the baselines may be systematically disadvantaged in Table 1 as well.
>
> 2. You state in your rebuttal that "SymmCD performs **significantly** better than prior works at generating crystals with realistic, diverse symmetries,". How do the authors come to the conclusion that their model performs significantly better without any statistical test or even crossvalidation?
>
> 3. The authors stated that crossvalidation is too costly. What are the cost exactly? Considering the computational cost that the authors themselves describe in Table 4 it appears quite reasonable to crossvalidate in my opinion, especially in light of the reproducibility questions.
>
> 4. The other papers e.g. FLowMM and DiffCSP++ report results on other datasets, e.g. Perov-5 and MPTS-52, Carbon-24 . Is there a reason why other datasets are not experimented with here? It appears to be the standard practice in the field. I am particularly curious since these datasets are part of the uploaded anonymous GitHub repo from the authors. Showing comprehensive results on multiple datasets would also be an option to strengthen the confidence in the empirical results.
>
> 5. Model selection. In your rebuttal you state "Next, we performed a random sweep of other architecture parameters, running 150 different hyperparameter combinations and choosing a model that had high performance on structural validity, compositional validity, and d_E" What do the authors mean by "a model"? In my opinion it would be good practice to define a specific model selection rule and choose all models in crossvalidation and hyperparameter search space for all models including baselines.
>
> I would be curious to hear what the  authors and other reviewers think about my concerns.

---

> > ### Author Response · Authors · 2024-11-23
> > **Response**
> >
> > We thank the reviewer for their response. We take their concerns seriously and answer each point below, (split into two comments):
> >
> > ### 1. Hyperparameter selection:
> >
> > To be clear, we have rerun the code provided for other methods using their reported optimal hyper-parameters, which they recommended for MP-20. For example, FlowMM reports a much more extensive hyper-parameter sweep based on optimizing the stability of structures after relaxing them with CHGNet (i.e. the metric we report in Table 3). Our hyperparameter optimization setup is more modest by comparison, and is therefore far less optimized than FlowMM.
> >
> > The reviewer is correct that our reproduction does not agree with their reported results (we sometimes get better and sometimes worse than what is reported), despite using their code and hyperparameters. We actually noticed this difference and got in touch with the authors of DiffSCP and DiffSCP++. They confirmed that they also observed fluctuations. This is why we did not make any claim in the paper regarding which method is superior based on these metrics. We would also be happy to report the results from the original papers, but these changes do not significantly affect our conclusions from Table 2. To the best of our knowledge, no prior research work on generative models for materials generation (including all the baselines) uses cross-validation for this task.
> >
> > ### 2. Significance:
> >
> > Our main result, for which we claim our method is substantially better, is the large difference in the distribution of space groups between the dataset distribution, the distribution obtained by SymmCD and the distribution obtained by all the other methods (except for DiffCSP++ for reasons we detail in the main text - this method directly used templates from the dataset). We only claim an improvement in the ability to generate crystals with realistic and diverse symmetries (not with respect to any other evaluation metric), which we think is clearly shown in Figure 4.
> >
> > As you rightly point out, we don’t want to suggest that our results are generally better in the sense of statistical significance. We would like to clarify that we never use the term “significantly better” when discussing our results in the manuscript, except for computational efficiency. For the proxy metrics, the only claim we make from these results compared to other methods is that “SymmCD performs on par with other methods across different metrics.”.
> >
> > The generation of crystals with realistic, diverse symmetries has been a neglected problem, despite being necessary in practical searches for materials with desirable physical properties. We focus our work on addressing this task. We additionally show that it can do so without compromising on other criteria.
> >
> > ### 3. Cross validation:
> >
> > All baselines on this task use the standardized data split between train, validation, and test. In the case of cross-validation, we would meaningfully depart from the established evaluation method in the literature.
> >
> > Furthermore, only the coverage and distance proxy metrics use the validation and test set (in Table 2); the space group distribution, templates, and stability metrics depend only on the trained model. Therefore, our primary evaluation settings and metrics don’t rely on a validation or test set. Also note that despite the efficiency of our model, cross-validation would involve retraining each model (not just ours) from scratch K times. **However, taking your feedback into consideration, we will retain each baseline and perform 5-fold cross-validation for the proxy metrics.** Unfortunately, this will not be feasible by the end of the discussion period and will take several weeks. Hence, we will include them in the camera-ready version of the paper if it is accepted.

---

> > > ### Author Response · Authors · 2024-11-23
> > > **Response (2/2)**
> > >
> > > ### 4. Other datasets:
> > > As our aim with SymmCD was to produce a single model that learns the general space of known stable materials for practical use in de novo generation, we focused our experiments on doing an extensive evaluation on MP-20. MP-20 is a much larger, more practically relevant, and more diverse dataset compared to Carbon or Perov. Carbon contains only all-carbon materials, and Perov only contains materials from a very small set of space groups (5 to be exact, whereas 169 are present in MP20), so neither is of as much interest to our goals. Our choice is similar to recent works, such as FlowMM, which only reports results on MP-20 for de-novo generation. We also note that none of the methods we compare against use MPTS-52 for de novo generation, only for crystal structure prediction (CSP), which is a separate task. As explained in the paper, CSP is not a relevant use case for our method.
> > >
> > > ### 5. Model selection:
> > >
> > > By “model” we mean a specific choice of architecture along with the hyperparameters. With respect to hyperparameter tuning and cross-validation for other methods, we believe we have answered this point above. Note that all of our practices reported in the paper are standard for the relevant literature.
> > >
> > > ---
> > > ---
> > > We would like to again thank the reviewer for raising their concerns, and appreciate their sense of rigour.
> > > We have made an effort to present our work as providing a solution grounded in crystallography theory to the important problem of generating crystals with non-trivial symmetries, as well as improving the computational efficiency of such generative models by leveraging symmetry. We have been careful in not making claims about the superiority of our method on other aspects.
> > >
> > > We are open to answering any other question the reviewer might have.

---

> > > > ### Author Response · Authors · 2024-11-30
> > > >
> > > > Dear Reviewer 7azh,
> > > > We again appreciate the feedback you have provided, it has been helpful in improving the reproducibility of our work and in clarifying its claims. We are curious to hear your thoughts on our responses to your concerns, and whether you have any further questions or feedback. As a gentle reminder, the last day for reviewer responses is December 2nd.

---

### Meta-Review · Area_Chair_jk99 · 2024-12-23

**Metareview:**

Summary
=======
The paper presents SymmCD, a diffusion-based generative model for creating crystal structures while preserving symmetry. The key innovation is the use of asymmetric units and site symmetry representation, combined with a binary matrix encoding for symmetry groups. The method achieves comparable performance to existing approaches but with improved computational efficiency. The authors evaluate their approach on structure prediction tasks and demonstrate the model's ability to generate diverse, valid crystal structures across different symmetry groups.

Strengths:
========
* Novel representation scheme that explicitly incorporates crystallographic symmetry
* Improved computational efficiency compared to existing methods, particularly in memory usage and training speed
* Well-structured and clear presentation, making complex crystallographic concepts accessible
* Successfully generates diverse and valid crystal structures across multiple symmetry groups

Weaknesses:
==========
* Limited performance gains compared to state-of-the-art methods
* Lack of code availability and reproducibility details
* Needs more details, e.g.,  hyperparameter, training algorithm
* Limited evaluation of physical properties (e.g., thermodynamic and mechanical stability)
* There was a major concern about reproducibility and comprehensiveness of evaluation.

Reasons for decision
================
The novel symmetry-preserving representation and improved computational efficiency represent valuable contributions to the field of crystal generation. This is not of a broad interest to ICLR audience as the theoretical analysis is rather thin. Note that this is not the first paper discussing group symmetry preserving in crystal generation, e.g., https://openreview.net/forum?id=dJuDv4MKLE. Nevertheless, the idea still represents useful advances when symmetry needs to be respected in generative modelling.

**Additional Comments On Reviewer Discussion:**

Two out of three reviewers explicitly stated they are not expert in crystal modelling, hence the discussions were a bit shallow.

---

### Decision · Program_Chairs · 2025-01-22

Accept (Poster)